# Molecular insight into RNA polymerase I promoter recognition and promoter melting

Yashar Sadian[1,3], Florence Baudin[1], Lucas Tafur[1,2,4], Brice Murciano[1], Rene Wetzel[1], Felix Weis[1] & Christoph W. Müller [1]*

RNA polymerase I (Pol I) assembles with core factor (CF) and Rrn3 on the rDNA core promoter for transcription initiation. Here, we report cryo-EM structures of closed, intermediate and open Pol I initiation complexes from 2.7 to 3.7 Å resolution to visualize Pol I promoter melting and to structurally and biochemically characterize the recognition mechanism of Pol I promoter DNA. In the closed complex, double-stranded DNA runs outside the DNA-binding cleft. Rotation of CF and upstream DNA with respect to Pol I and Rrn3 results in the spontaneous loading and opening of the promoter followed by cleft closure and positioning of the Pol I A49 tandem winged helix domain (tWH) onto DNA. Conformational rearrangement of A49 tWH leads to a clash with Rrn3 to initiate complex disassembly and promoter escape. Comprehensive insight into the Pol I transcription initiation cycle allows comparisons with promoter opening by Pol II and Pol III.

[1] European Molecular Biology Laboratory (EMBL), Structural and Computational Biology Unit, Meyerhofstrasse 1, 69117 Heidelberg, Germany. [2] Collaboration for joint PhD degree between EMBL and Heidelberg University, Faculty of Biosciences, 69120 Heidelberg, Germany. [3] Present address: Bioimaging Center, University of Geneva, 30, Quai Ernest-Ansermet 4, CH-1211 Geneva, Switzerland. [4] Present address: Department of Molecular Biology, University of Geneva Sciences III, 30, Quai Ernest-Ansermet 4, CH-1211 Geneva, Switzerland. *email: cmueller@embl.de

Transcription of eukaryotic genes requires the recruitment of RNA polymerase to promoter DNA through general transcription factors that assemble into a complex with double-stranded DNA (dsDNA) known as closed complex (CC). Subsequently, melting of the DNA duplex around the transcription start site (TSS) results in an open complex (OC). RNA synthesis marks the transition into the initially transcribing complex (ITC), which undergoes further structural rearrangements to form an elongation complex (EC) once productive RNA synthesis is established and RNA polymerase has escaped from the promoter[1–3]. Recent studies have identified promoter meltability of the 'initially melted DNA region' upstream of the TSS as an important parameter that allows opening of Pol I and Pol III promoters, but also in a subset of Pol II promoters, without ATP-dependent translocases such as TFIIH[4].

Transcription of pre-ribosomal RNA initiates by binding of RNA polymerase I (Pol I) to Rrn3, which stabilizes its monomeric form[5–7]. Subsequently, Pol I and Rrn3 are recruited to the core promoter by the upstream activating factor (UAF), TATA-binding protein (TBP) and core factor (CF) to form the Pol I pre-initiation complex (PIC). However, promoter-dependent transcription in vitro only requires Pol I, Rrn3 and CF[8]. Recent studies have shown that Pol I initiation complexes (IC) are structurally different from Pol II and Pol III ICs[9–11]. In the Pol I ICs, the upstream DNA occupies a different position and the TFIIB-related CF subunit Rrn7 binds further upstream from Pol I compared to the Pol II and Pol III ICs[1,2,12,13]. In comparison, the three Pol I IC structures show similar overall architectures, similar DNA conformations and CF-DNA interactions[9–11]. Differences include the presence of RNA in one IC reconstruction[9], the presence of the A12.2 C-terminal domain in the active site concomitant with a slightly more open DNA-binding cleft in another IC reconstruction[11] and finally the absence of Rrn3 in the third reconstruction[10]. A more detailed comparison is provided in a recent review[14].

Structural modelling of the rDNA promoter region from yeast and other organisms has predicted a bend and sequence-dependent flexibility in the upstream region of the DNA, important for promoter recognition[9,15]. Randomization of the promoter sequence also showed that the region from base pair (bp) −30 to −1 is critical for in vitro transcription[9] consistent with the observation that CF recognizes the promoter from bp −27 to −16 (ref. [10]). However, the limited resolution of the Pol I IC structures[9–11] and lack of complementary biochemical data precluded a more detailed understanding of how CF subunits recognize promoter DNA and facilitate DNA opening in an ATP-independent manner. Moreover, the available Pol I IC structures did not provide a complete understanding of the structural rearrangements during IC assembly, promoter DNA opening and promoter escape.

Here, we present cryo-electron microscopy (cryo-EM) reconstructions of several Pol I ICs, including two CCs, two spontaneously formed OC intermediates and two OCs varying between 2.7 to 3.7 Å resolution. Together, the different reconstructions provide mechanistic insights into complex assembly, allow following template and non-template DNA strand paths, show the necessary conformational changes during promoter opening and reveal the interplay between Rrn3 and A49 tWH in promoter escape. In addition, the structural data, combined with biochemical experiments, also allow better understanding the specificity of promoter recognition by CF and A49 tWH.

## Results

### Cryo-EM structure determination of Pol I CC and Pol I OC.
To assemble Pol I ICs, we used a double-stranded DNA scaffold containing the *Saccharomyces cerevisae* core rDNA promoter from bp −50 to +20 (with +1 denoting the TSS) to prepare the Pol I CC, and a similar transcription scaffold with the same sequence but containing a 15-nucleotide (nt) mismatch (bp −10 to +5) to prepare an artificially induced Pol I OC (Methods). Initial particle classification revealed that, as in Pol II (ref. [1]) and Pol III (ref. [13]), most of the particles from the Pol I CC dataset underwent spontaneous DNA opening and therefore represented OCs. Nevertheless, we obtained a CC reconstruction (CC1) at 3.7 Å resolution (Table 1, Supplementary Figs. 1 and 4). To prevent spontaneous DNA opening and increase the proportion of CC particles in the sample, we reduced incubation time of the complex by mixing all components immediately before plunge freezing (Supplementary Fig. 2). The majority of particles again represented OCs, but we also obtained a class with closed DNA and good CF-DNA density (CC2) at an overall resolution of 2.9 Å (Table 1, Supplementary Figs. 2 and 4). From the Pol I OC sample, we obtained two classes at 3.5 Å (OC1) and 3 Å resolution (OC2), where the OC2 revealed additional density for the A49 tWH at a previously unobserved position, while in the OC1 the A49 tWH is disordered (Table 1, Supplementary Figs. 3 and 4). Multibody refinement on Pol I-Rrn3 and CF densities in the OC2 improved the local resolution to 2.9 and 3 Å for Pol I-Rrn3 and CF, respectively (Supplementary Fig. 4). The improved resolution in the CF-upstream DNA region allowed us to better describe specific interactions between CF and the A49 tWH with promoter DNA (see below).

### Structure of the Pol I CC.
Comparison of the Pol I CCs with the previously published Pol I ICs reveals that CF and the upstream DNA bound by CF both occupy roughly similar positions with respect to Pol I-Rrn3 (refs. [9–11]). In contrast, the upstream DNA preceding the TSS adopts an unexpected position by being strongly bent away from the DNA-binding cleft at bp −11 to −7 at an angle of ~60° (Fig. 1a). CC1 and CC2 differ in the position of CF and upstream DNA with respect to Pol I-Rrn3 with CC1 being rotated out by ~15° compared to CC2. CF and upstream DNA bound by CF pivot around the TSS region, which is held in the same place relative to Pol I-Rrn3 in CC1 and CC2 (Fig. 1b). In CC2, CF adopts a position more similar to the previous IC structures[9–11] and appears to be less mobile than in CC1. As proposed previously, extension of the dsDNA in the Pol I IC models[9–11] would clash with the clamp, rudder and Rpb5 (ref. [9]). This clash is avoided by a notable kink at bp −11 to −7 in the Pol I CCs, which distorts the double helix and projects the DNA away from the Pol I DNA-binding cleft (Fig. 1c). In the Pol I CCs, the DNA-binding cleft is occupied by the DNA-mimicking loop (DML)/expander (Fig. 1c) as in the crystal structure of apo Pol I (refs. [16,17]). The DML/expander in the cleft can only be accommodated because Pol I adopts an open clamp conformation most similar to the monomeric apo Pol I and Pol I-Rrn3 structures[5–7] (cleft width: 36 Å in Pol I CC; 40 Å in apo Pol I crystal structure, Supplementary Fig. 5). The DML/expander interacts with the partially unfolded bridge helix as shown in the Pol I crystal structure, which stabilizes the open cleft conformation[16]. An open cleft conformation is further promoted by the stable anchoring of the A12.2 C-terminal domain in its TFIIS-like position as observed in the previous Pol I PIC[11].

The open DNA-binding cleft allows Pol I to accommodate downstream dsDNA between the clamp and the lobe where it is maintained by interactions with the protrusion, rudder and the clamp coiled-coil that is formed by the largest subunit A190 at the basis of the clamp (A190 residues 380–470). Furthermore, the DNA in the Pol I CC is excluded from the cleft through steric hindrance and electrostatic repulsion by the negatively charged DML/expander. Upon cleft closure and transition to the OC

**Table 1 Cryo-EM data collection, refinement and validation statistics.**

| | CC1 EMD-4982 PDB 6RQH | CC2 EMDB-4984 PDB 6RQL | OC1 EMDB-10007 PDB 6RUO | OC2 EMDB-10038 PDB 6RWE | Intermediate 1 EMDB-4987 PDB 6RRD | Intermediate 2 EMDB-10006 PDB 6RUI | Pol-Rrn3-tWH EMDB-4985 PDB 6RQT |
|---|---|---|---|---|---|---|---|
| **Data collection and processing** | | | | | | | |
| Magnification | 135,000 | 135,000 | 105,000 | 105,000 | 135,000 | 135,000 | 105,000 |
| Voltage (kV) | 300 | 300 | 300 | 300 | 300 | 300 | 300 |
| Electron exposure (e$^-$/Å$^2$) | 42 | 44 | 62 | 62 | 42 | 44 | 62 |
| Defocus range (μm) | −0.75 to −3 | −0.5 to −2.5 | −0.75 to −3 | −0.75 to −3 | −0.75 to −3 | −0.5 to −2.5 | −0.75 to −3 |
| Pixel size (Å) | 1.04 | 1.04 | 1.32 | 1.32 | 1.04 | 1.04 | 1.32 |
| Symmetry imposed | No | No | No | No | No | No | No |
| Initial particle images (n) | 1,347,851 | 5,148,501 | 1,988,127 | 1,988,127 | 1,347,851 | 5,148,501 | 1,988,127 |
| Final particle images (n) | 9483 | 24,482 | 75,851 | 59,963 | 24,848 | 42,727 | 9789 |
| Map resolution (Å) (FSC = 0.143) | | | | | | | |
| Average global resolution (Å) | 3.76 | 2.92 | 3.50 | 3.03 | 3.11 | 2.72 | 4.02 |
| Average CF-focused resolution (Å) | 9.69 | 2.99 | 3.08 | 3.08 | 3.32 | 2.82 | — |
| Average Pol-focused resolution (Å) | 3.66 | 2.88 | 2.96 | 2.96 | 3.06 | 2.67 | — |
| Map resolution range (Å) | 3.5 to 10 | 2.8 to 5 | 3.2 to 7.5 | 2.8 to 5.5 | 2.9 to 6 | 2.6 to 5.8 | 3.8 to 8.5 |
| **Refinement** | | | | | | | |
| Initial model used (PDB code) | 6RQH | 6RQL | 6RUO | 6RWE | 6RRD | 6RUI | 6RQT |
| Model resolution (Å) (FSC = 0.5) | 4.0 | 3.3 | 4.0 | 3.2 | 3.3 | 3.1 | 4.4 |
| Map sharpening B factor (Å$^2$) | −65.6 | −39.4 | −64.8 | −45.1 | −47.9 | −34.8 | −94.3 |
| Model composition | | | | | | | |
| Non-hydrogen atoms | 50,851 | 51,627 | 50,792 | 53,719 | 51,631 | 51,525 | 39,820 |
| Protein residues | 6151 | 6246 | 6086 | 6454 | 6179 | 6179 | 4915 |
| Ligands | N/A | N/A | N/A | N/A | N/A | N/A | N/A |
| B factors (Å$^2$) | | | | | | | |
| Protein | 127.8 | 86.5 | 132.9 | 98.4 | 33.8 | 81.5 | 197.6 |
| Ligand | N/A | N/A | N/A | N/A | N/A | N/A | N/A |
| R.m.s. deviations | | | | | | | |
| Bond lengths (Å) | 0.0081 | 0.0094 | 0.0077 | 0.0093 | 0.0085 | 0.0106 | 0.0076 |
| Bond angles (°) | 1.06 | 1.08 | 1.04 | 1.016 | 1.083 | 1.089 | 1.189 |
| Validation | | | | | | | |
| MolProbity score | 2.0 | 1.86 | 2.0 | 1.84 | 1.83 | 1.82 | 2.41 |
| Clashscore | 8.0 | 6.6 | 7.91 | 6.72 | 6.14 | 6.57 | 13.78 |
| Poor rotamers (%) | 0.58 | 0.74 | 0.38 | 0.73 | 0.38 | 0.81 | 1.37 |
| Ramachandran plot | | | | | | | |
| Favoured (%) | 89.35 | 91.7 | 89.43 | 92.49 | 91.82 | 92.65 | 85.02 |
| Allowed (%) | 10.50 | 8.28 | 10.52 | 7.43 | 8.13 | 7.3 | 14.84 |
| Disallowed (%) | 0.15 | 0.02 | 0.05 | 0.08 | 0.05 | 0.05 | 0.14 |

conformation (see below), the clamp coiled-coil would clash with the dsDNA in the CC conformation; therefore, a closed clamp is incompatible with the CC conformation (Supplementary Fig. 6).

The higher resolution structures of the Pol I CCs and OCs also help to elucidate the role of Rrn3 and its interaction with CF. Rrn3 has been previously suggested to stabilize the monomeric conformation of Pol I[5,6] and previous biochemical experiments showed that Rrn3 is absolutely required for promoter-dependent transcription and formation of a productive Pol I PIC[6,18]. Likewise, Rrn3 appears to be required for the formation of a productive PIC in the mammalian system[19]. However, none of the previous structures detected any direct interactions between CF and Rrn3. In the Pol I CC, Rrn3 is stably bound to Pol I as in the structure of Pol I-Rrn3 without DNA[5,6]. Interestingly, our CC2 reconstruction reveals a large fragment of an acidic loop of Rrn3 (residues 245–325), which has neither been observed in the Rrn3 crystal structure[18] nor in the previous cryo-EM reconstructions[5,6,9,11] (Fig. 1d). This 80-residue loop contains many negatively charged residues and folds onto the Rrn3 surface, close to a region of Rrn3 enriched with positively charged residues. Most importantly, the acidic loop wraps around the Rrn7 Zn-ribbon, 'closing' the Pol I dock domain and stabilizing the interaction of the CF with Pol I. Effectively, the acidic loop of Rrn3 appears to be the only contact point that connects Pol I, CF and Rrn3.

**Pol I OC2 reveals a TFIIE-like position of the A49 tWH.** The overall architecture of the Pol I OC1 at 3.5 Å resolution and the Pol I OC2 at 3.0 Å resolution are very similar. However, important differences comprise a more closed clamp and the A49 tWH stably bound to the upstream DNA in the OC2, while it is mobile in the OC1 (Fig. 2a, b; Supplementary Fig. 7 shows a direct comparison between OC1 and OC2). The DNA-binding cleft in the OC2 is slightly narrower compared to the OC1 because of the movement of Pol I modules 1 and 2 (refs. [16,20]) towards each other (Fig. 2c). The movement of the two modules in both Pol I OC structures results in an intermediate clamp conformation between the Pol I CC (more open) and the Pol I EC (more closed)[20,21], confirming that progressive clamp closure is a feature of transcription initiation in Pol I. The clamp conformation in the OC2 allows the A49 linker to wrap around the clamp coiled-coil and to position the C-terminal A49 tWH close to the upstream DNA (Fig. 2c), where it interacts (together with CF) with the promoter DNA (see below). Interestingly, this TFIIE-like position of the A49 tWH (also observed in the

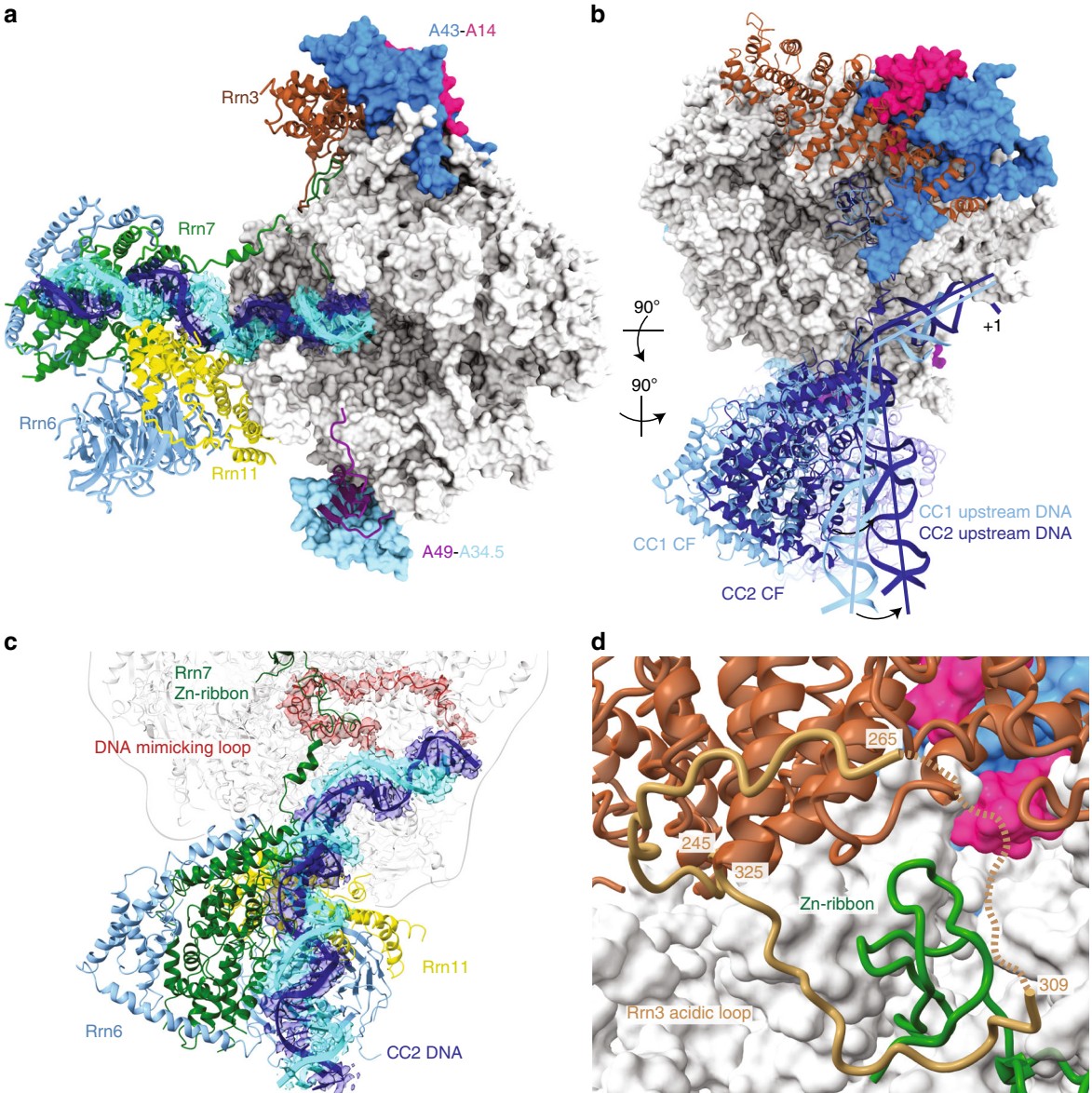

**Fig. 1 Structure of the Pol I CC. a** The structure of the Pol I CC2 is depicted with Pol I in light grey and the stalk subunits, A43 and A14, in blue and violet, respectively. Rrn6, Rrn11, Rrn7, Rrn3 and Pol I A49 are shown in light blue, yellow, dark green, brown and purple ribbons, respectively. The template and non-template DNA strands and the corresponding density from the sharpened globally refined maps are shown in dark blue and cyan, respectively. **b** Transition between the CC1 and CC2 states. The structures were superimposed onto subunit Rpb8. A line shows the direction of upstream and downstream DNA duplexes in both complexes. **c** The downstream DNA is kinked at bp −11 to −7 and excluded from the Pol I cleft by the DML/expander. Pol I CC2 is shown in ribbon representation with colours as in **a**. Cryo-EM density for the template and non-template DNA strands and the DNA-mimicking loop (DML) obtained from the sharpened multibody-refined maps are shown in dark blue, cyan and red, respectively. **d** The acidic loop of Rrn3 (residues 245–325) of Pol I CC2 contributes to the stabilization of the Rrn7 Zn-ribbon (dark green). Rrn3 is coloured in brown and the acidic loop in dark khaki.

spontaneously formed OC) differs from its position in the EC structure[20] and the EC-like position in the Pol I ITC lacking Rrn3[10] and in this position, A49 tWH would clash with the upstream DNA in the EC (Fig. 2d).

The position of A49 tWH in the OC2 is similar to the position of yeast Tfa2 (human TFIIEβ) WH1/WH2 in the Pol II PIC[1] (Supplementary Fig. 8). TFIIE has been shown to be required for DNA opening in the Pol II IC[1]. The A49 tWH was suggested to be homologous to TFIIEβ (yeast Tfa2)[6,22] and plays a role in promoter-specific transcription in vitro[6] and might therefore play a similar role as TFIIE in promoter recognition and DNA opening during Pol I initiation. Moreover, the A49 dimerization and linker domains are homologous to TFIIEα (Tfg1) in the Pol II system[22]. The A49 linker contains a charged helix (residues

130–150), similar to the TFIIEα (Tfg1) charged helix, and both localize next to the non-template DNA strand. Additionally, positively charged residues in the A49 linker point towards the cleft and appear to constrain the path of the non-template strand, which is not entirely resolved but can only be accommodated between the fork loop 1 (FL1) (A135, residues 470–484) and loop A (A135, residues 260–271) (Supplementary Fig. 10a, b). Transition of the A49 tWH from the OC2 position to the EC position[20] involves an ~30 Å movement and rotation towards the stalk, both of which are facilitated by the flexible A49 linker region (residues 170–190) (Supplementary Fig. 9a). Interestingly, the A49 tWH position in the EC clashes with Rrn3, explaining why Rrn3 was missing in the ITC structure when A49 tWH was resolved in the EC position[10] (Supplementary Fig. 9b).

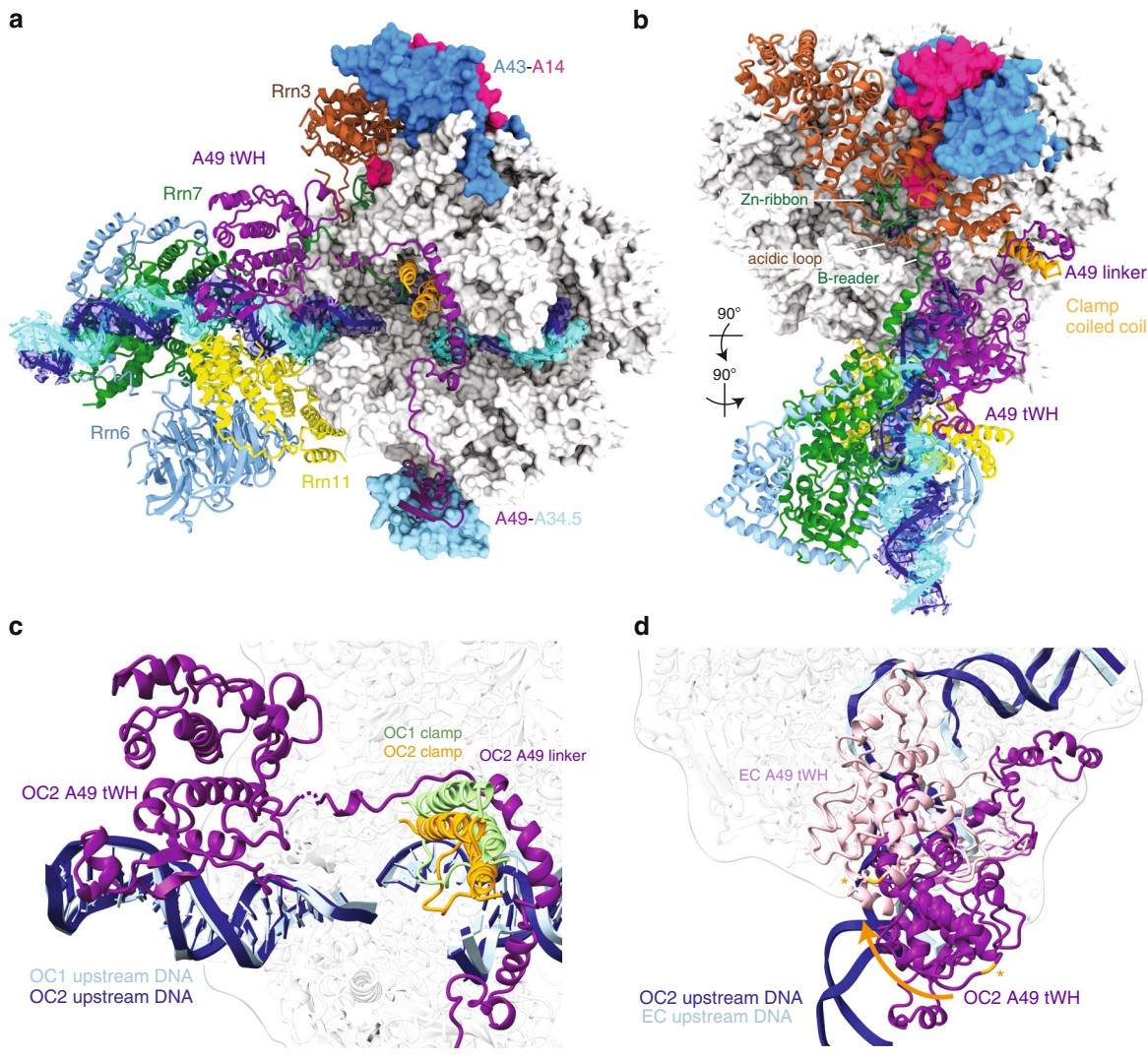

**Fig. 2 Structure of the Pol I OC with A49 tWH contacting promoter DNA. a** The structure of the Pol I OC2 is coloured as in Fig. 1a. The clamp coiled-coil is highlighted in light orange. Cryo-EM density of the sharpened globally refined maps is shown for both DNA strands. **b** Top view of the OC2. **c** The clamp is more closed in the OC2 (light orange) than in the OC1 (light green) allowing the A49 linker to wrap around the coil-coil and A49 tWH to contact the promoter. Models were aligned on subunit A135. **d** The position of the upstream DNA (dark blue) and A49 tWH in the OC2 (dark purple) is compared with that in the Pol I EC (upstream DNA in light blue (PDB: 5M5X) [http://dx.doi.org/10.2210/pdb5m5x/pdb] and A49 tWH in light pink (PDB: 5M64) [http://dx.doi.org/10.2210/pdb5m64/pdb]). The movement of the A49 tWH is indicated by the orange arrow. The asterisks indicate residue Q252 in the OC2 and EC. Structures are aligned on subunit A135 as in **c**.

We obtained an independent reconstruction of the Pol I-Rrn3-DNA complex from the OC dataset lacking the CF where the A49 tWH is observed in the same position as in the Pol I EC complex[20,21] (Supplementary Figs. 3 and 9b). The movement of the A49 tWH towards the stalk appears to destabilize the interaction of Rrn3 with Pol I subunit A43 by pushing the Rrn3 N-terminal region away from the stalk. It was previously suggested that A49 is involved in the release of Rrn3 during the early phase of elongation[23–28]. However, no mechanistic insight of how A49 could regulate the release of Rrn3 was available. Our results thus provide structural evidence that rationalize the previous observations on the role of the A49 tWH in promoter opening and Rrn3 dissociation.

**DNA opening during Pol I initiation**. In addition to OC1 and OC2, we also obtained several spontaneously formed OC reconstructions from the CC1 and CC2 datasets that differ in the size of the transcription bubble. Focused classification on the downstream DNA resulted in several reconstructions with different degrees of DNA opening and clamp conformations (Fig. 3). However, the density for the single-stranded DNA in the DNA-binding cleft was fragmented, indicating heterogeneity in the size of the transcription bubbles. The resulting models show progressive DNA loading and melting around the TSS. In the first intermediate (OC Intermediate 1), the DNA is loaded and trapped in the active site by closure of the clamp, which in concert with the rudder, promotes the formation of a short bubble. In this intermediate, the non-template strand is still next to the template strand. Subsequently, a slight opening of the clamp allows extension of the transcription bubble and exclusion of the non-template strand from the active site (OC Intermediate 2). Finally, in the OC, the transcription bubble opens to ~11–15 nucleotides (OC2), the rudder is inserted between the template and non-template strands to prevent re-annealing of both strands, and the template strand is positioned correctly for initial RNA synthesis.

Compared to TFIIE in Pol II and C82 in Pol III, there is no element in the A49 tWH equivalent to the 'E-wing' (TFIIE) or

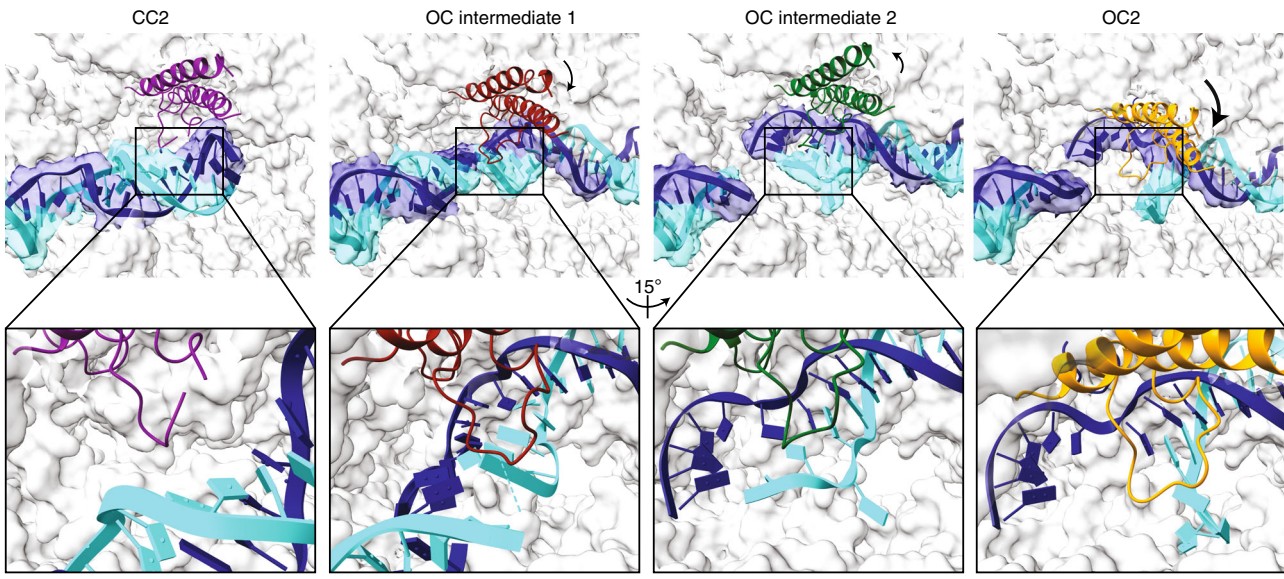

**Fig. 3 Mechanism of DNA loading/melting.** Pol I CC2, Pol I OC intermediate 1, Pol I OC intermediate 2 and Pol I OC2 are depicted from left to right with the rudder (A190 residues 443–455) shown in purple, red, green and orange ribbons. The cryo-EM density for the DNA from 6 Å low-passed filtered maps is shown in the upper panel; zoomed in and rotated views of the upper panel are shown in the lower panel. Initial DNA loading and formation of a small transcription bubble requires clamp closure, while further DNA promoter melting requires clamp opening and insertion of the rudder between the template and non-template strands.

'cleft loop' (C82)[1,13] (Supplementary Fig. 8) that inserts between the two strands of the DNA duplex and thereby causes DNA opening. Instead, the Pol I rudder might fulfil a similar role. Interestingly, the Rrn7 B-reader, which in both, Pol I CC and OC, inserts into the RNA exit channel, interacts with both the lid loop and the rudder. As these elements are directly linked to the clamp, the Rrn7 B-reader might influence DNA opening by modulating the movement of the clamp, lid loop and rudder. DNA opening and loading of the template strand in Pol I thus might parallel the mechanisms proposed in Pol II and Pol III, which is mediated by movements of the clamp and an extended loop (rudder in Pol I, E-wing in Pol II and C82 cleft loop in Pol III)[1,13].

The mechanism of promoter melting has been extensively studied in bacterial RNA polymerases[29,30]. Transcription bubble formation initiates with the recognition of base −12 followed by base flipping at −11 that induces strand separation. Base flipping at both ends of the bubble (−11 and +2) seems to stabilize the bubble by preventing the re-annealing of the strands[31,32]. Interestingly, our OC reconstructions (OC Intermediate 2 and OC2) also show flipping of the +2 base in the non-template strand as in the bacterial RNA polymerase IC[32] and in the nucleotide-analogue-bound Pol I ECs[33]. In addition, we also observe a similar positioning of the +1 base as in the Pol I ECs, suggesting that interactions with bases +2 and +1 serve to stabilize the transcription bubble already in the OC state. Although the upstream edge of the bubble is not well resolved, it is located close to the Pol I wall and protrusion as previously observed[10,11]. Interestingly, the Rrn7 B-linker and the A49 linker are positioned close to the edge of the transcription bubble and contain several basic residues to stabilize upstream DNA during the initial stage of transcription bubble formation (Supplementary Fig. 10a, c).

**Mechanism of promoter recognition by CF and the A49 tWH.** Focused refinement on CF and upstream DNA of CC2 and OC2 resulted in cryo-EM maps with improved side chain and nucleotide densities compared to previous Pol IC structures[9–11] and allowed a more complete description of CF residues contacting DNA, as well as the interactions between the A49 tWH

and promoter DNA (Fig. 4). Together, CF subunits Rrn7 and Rrn11, and A49 tWH completely encircle the promoter DNA similarly to TFIIIB in the Pol III PIC[12,13].

Interaction of CF with DNA is mediated via several elements of the N- and C-terminal cyclin folds of Rrn7, and the TPR protein Rrn11, most of which were previously shown to be involved in DNA interaction and complex formation[10]. However, the higher resolution of the OC2 reconstruction allowed us identifying the register of the amino acid residues that specifically interact with promoter DNA as a basis for subsequent mutational studies. In Rrn7, the main elements include the loop between helices α2 and α3 and the extended loop between helices α4 and α5 in the N-terminal cyclin domain and a loop between α7 and α8 in the C-terminal cyclin domain. In Rrn11, the N-terminal DNA-binding helical bundle (DBHB)[11] and the loop between helices α8 and α9 contact DNA in the major and minor groove, respectively (Fig. 4a–f). The DNA-interacting residues in Rrn7 and Rrn11 mainly contact the backbone of both DNA strands (bp −32 to −13), although several residues also contact DNA bases. In Rrn7, residues N209, R293 and H294 contact DNA bases adenine in bp −20, guanine in bp −26 and adenine in bp −27, respectively. In particular, R293 deeply inserts between the DNA bases of bp −27 and bp −26 (Fig. 4d). In Rrn11, only residue R11 makes base-specific contacts with two adjacent adenines in bp −21 and bp −22 (Fig. 4e, i). Notably in its TFIIE-like position, A49 tWH interacts with both DNA strands from bp −25 to −17 via residues K356 and S358 and loop residues K386, S387, S391 and K393, some of which were already shown to be important for binding of A49 tWH to the Pol I promoter[34] (Fig. 4g, i). Together the multiple DNA interactions mediated by the N- and C-terminal cyclin folds of Rrn7, the DBHB of Rrn11 and the A49 tWH induce considerable bending and distortion of the promoter DNA (Fig. 4a). The major groove is considerably widened from bp −27 to −22 where the C-terminal cyclin fold of Rrn7 and the DBHB of Rrn11 contact DNA bases and backbone in the major groove, while from bp −24 to −19 the minor groove is widened and resembles the groove width of A-DNA (Fig. 4a, c, d).

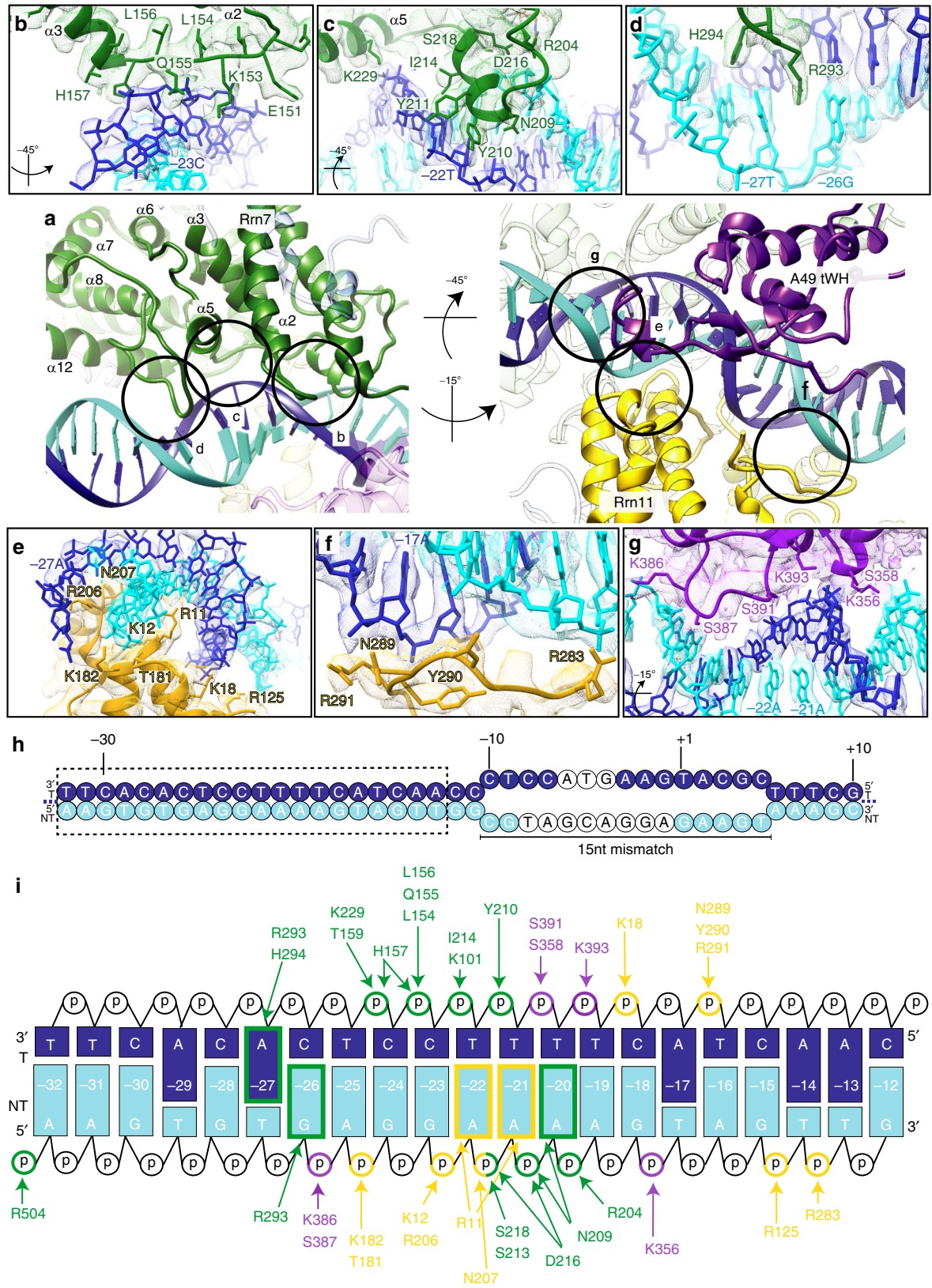

**Specificity of CF binding and Pol I transcription**. It was previously shown that 10 bp sequence randomization in the −30 to +3 region abolishes promoter-dependent transcription in vitro[9]. However, the sequence specificity of CF binding and the critical bases for transcription were not characterized. We therefore explored the specificity of CF binding using Exonuclease III

(ExoIII) footprinting experiments with radiolabelled double-stranded core promoter DNA (bp −50 to +20). ExoIII treatment on double-stranded promoter DNA bound to CF revealed two distinct bands at bp −32 on the template strand (delimiting the upstream DNA-binding boundary) and bp −9 on the non-template strand (delimiting the downstream DNA-binding

**Fig. 4 Promoter recognition by CF and A49 tWH. a** Interactions of Rrn7 (left panel) and Rrn11 and A49 tWH (right panel) with promoter DNA. Circled regions highlight the interactions that are shown in detail in subsequent panels **b** to **g**. The relative view with respect to panel **a** is indicated if different. The cryo-EM densities depicted in **b–g** were obtained from the sharpened, amplitude-corrected, multibody-refined (CF-upstream DNA) OC2 map. **b** The Rrn7 N-terminal cyclin fold contacts the upstream DNA at positions −25 and −24 with residues L154, Q155, L156, H157 in the loop between α2 and α3 (helices are denoted as in ref. [11]. **c** The Rrn7 N-terminal cyclin fold contacts the upstream DNA at position −25 to −19 with residues R204, N209, Y210, S213, I214, D216, S218 and K229 in the loop between α4 and α5. **d** The Rrn7 C-terminal cyclin fold contacts the upstream DNA at position −27 and −26 with residues R293 and H294 in the loop between α7 and α8. **e** Rrn11 contacts the upstream DNA at position −24 to −13 with residues R11, K12, K18, R125, T181 and K182 of the DBHB, and residues R206 and N207 of the TPR domain. **f** Rrn11 contacts the upstream DNA at −17 to −13 via R283, N289, Y290 and R291 in the loop between α8 and α9. **g** A49 tWH contacts the upstream DNA at position −25 to −17 via K356, S358, K386, S387, S391 and K393. **h** Schematic representation of the transcription scaffold used to prepare the artificially induced Pol I OC. The bases, which were not resolved in the final reconstruction of the Pol I OC2, are depicted as white circles. The dotted box indicates the binding area of CF on the core promoter (−32 to −13) as shown in **i**. **i** Detailed interactions of CF and A49 tWH with the specificity box. The pyrimidine and purine bases are depicted as small and large rectangles, respectively, and the phosphate backbone as white circles. The contact points are indicated with arrows either to the phosphate backbone/sugar or to the bases, which are coloured according to the colour of the interacting protein subunit.

boundary) demonstrating that binding of CF to the promoter DNA is highly specific (Fig. 5a). If we moved the CF-bound DNA sequence (bp −30 to −12) either five nucleotides downstream (block exchange 1, BE1) or six nucleotides upstream (block exchange 2, BE2), the CF footprint changed accordingly, highlighting that this region contains all the required elements to bind specifically to CF (Fig. 5a). Hence, we call this region the 'specificity box'. In the next experiments, we exchanged the DNA sequence in bp −27 to −23 (ACTCC to GAGAT) and in bp −22 to −19 (AAAA to GAGC) within the AT-rich region. Both resulted in loss of the specific CF footprint pattern (Fig. 5a). The observed CF-DNA interactions and footprinting experiments (Fig. 4i and 5a) are overall in good agreement with the results of Jackobel et al.[35] who used DNA competitor oligonucleotides to map the minimal, sequence-specific CF binding region from bp −28 to bp −17.

To assess the role of specific nucleotides present in the CF-binding site for the formation of a transcriptionally active complex, we performed in vitro promoter-dependent transcription assays (Fig. 5b). No transcription product is observed when either Rrn3 or CF is omitted from the reaction as observed before[6,8]. When the specificity box is moved (BE1 or BE2), transcription is completely abolished (Fig. 5b, top panel), although CF binds specifically (Fig. 5a). When bp −11 to −7 (CCTCC) was changed into (TTGAA), no transcription occurred in the wt sequence highlighting that the sequence of bp −11 to −7 is required for effective transcription (Fig. 5b, top panel). Interestingly, this sequence corresponds to the strongly bent region of the DNA in the Pol I CC (Figs. 1c and 5b, bottom panel).

Inside the specificity box, as expected, mutations of bp −29 to −27, −27 to −23, −24 to −23 or −22 to −19 did not produce any transcripts due to impaired specific CF binding (Fig. 5b, top panel). Single mutation −27C or double mutation −27C/−26G decrease the in vitro activity by roughly 90%, in contrast to mutation −25G, which has no effect on transcription and is not directly contacted by CF (see Figs. 4c and 5b, top panel). These results highlight the importance of the bases located in the specificity box for a specific recognition by CF. When we mutated Rrn7 residues 291–294 that directly contact bp −27/−26 all into alanine, we observed a reduced transcriptional activity that was even more pronounced when the charges were reversed (DERH into RHDE). Similarly in Rrn11, when we changed residues R11, K12, K16 and K18 all into alanine or glutamate, we also observed a severe decrease in Pol I transcriptional activity, showing the importance of these residues in Pol I promoter recognition (Fig. 5b, middle panel).

The C-terminal moiety of Rrn6 is not visible in the EM structure. To track its putative implication in transcription initiation, a CF mutant where the C-terminal moiety of Rrn6 (Δ776–894) is lacking

was used in an in vitro transcription assay. Compared to the wt CF, no change was observed in the in vitro Pol I activity (Fig. 5b) suggesting that this domain is not involved in CF/DNA recognition and initiation, which agrees with the fact that deletion of the C-terminal domain of Rrn6 did not affect yeast growth[36].

Deletion of the Rrn3 acidic loop (Δ253–319) resulted in a significant reduction of transcription compared to wt Rrn3 (Fig. 5b, middle panel). This highlights its functional role during Pol I initiation/early transcription. When the RNA reaches a critical length, it clashes with the Rrn7 B-finger, which then displaces the Rrn3 acidic loop. The absence of the acidic loop presumably prevents CF to dissociate from the complex and leads to poor transcription efficiency.

## Discussion

Here, we present previously uncharacterized structures of Pol I CCs, Pol I OC intermediates and Pol I OCs combined with biochemical data that further rationalize promoter recognition by CF and Pol I. Our structures illustrate the major structural rearrangements that result in spontaneous DNA opening and OC formation during Pol I transcription initiation (Supplementary Movie 1), thereby further completing the 'ratcheting' model proposed for Pol I transcription initiation[10]. In this model, movement of CF and CF-bound upstream DNA relative to Pol I generates strain on the DNA, which contributes to DNA melting. The ratcheting mechanism of CF-upstream DNA was initially suggested based on different OC conformations[10] and these were observed only in absence of the essential factor Rrn3. We now demonstrate that the ratcheting movement also occurs in the presence of all essential transcription factors and during the transition from the CCs to OCs, thereby further clarifying and corroborating this mechanism.

Our structures also highlight similarities and differences compared to transcription initiation by Pol II and Pol III (Fig. 6, Supplementary Movie 1). Promoter recognition by CF merely depends on DNA shape and DNA sequence-dependent deformability[9,15]. Our structures indeed show that the conformation of Pol I promoter DNA considerably deviates from standard B-DNA that includes considerable DNA bending and widening of the minor groove induced by CF and A49 tWH binding. In addition, the DNA duplex downstream of the CF binding site (bp −11 to −7) is also strongly bent in the CCs and melts during OC formation rationalizing the importance of this stretch of DNA for transcription initiation.

The Pol I-Rrn3 complex is recruited to the promoter already bound by CF to form the Pol I CC1 where CF is in a distal position (compared to Pol I ITC). At this stage, the DML/expander and A12.2C are present in the active site, which stabilize the Pol I open cleft conformation (Fig. 6a) and allows Pol I to accommodate

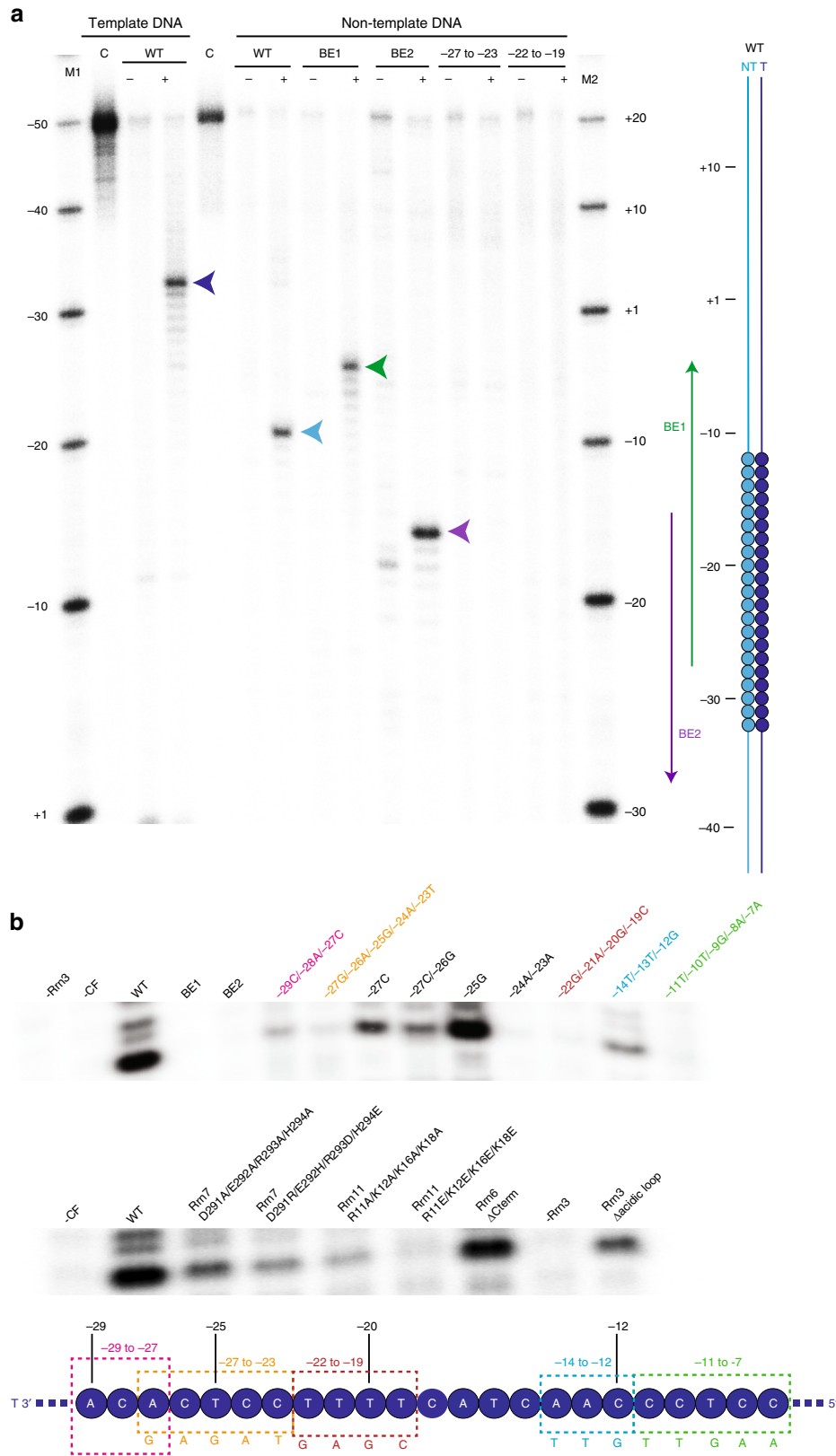

dsDNA through interactions with the clamp coiled-coil. The rotation of CF-upstream DNA with respect to Pol I-Rrn3 results in formation of the Pol I CC2 (Fig. 6b). Interactions between Rrn3, Pol I and CF are mediated through an 80-residue acidic loop in Rrn3 that fastens the Zn-ribbon of Rrn7 onto Pol I. Interestingly, the Pol I CC structures resemble the yeast Pol III CC where the DNA is kinked away from the cleft through interactions with the clamp coiled-coil[13]. The transition between the two Pol I CCs is also reminiscent of the Pol III CCs where the heterotrimer and the upstream DNA move away from the Pol III[13].

**Fig. 5 Biochemical characterization of CF promoter recognition. a** ExoIII footprinting assay performed on the dsDNA transcription scaffold (−50 to +20) labelled either on the template or non-template strand. Lanes from left to right: M1, marker for template strand; C, the transcription scaffold without ExoIII as a control; WT, wild-type (wt) transcription scaffold labelled on the template strand ± CF; C, the wt transcription scaffold labelled on the non-template strand without ExoIII as a control; samples of ±CF, the block exchange 1 (BE1) labelled on the non-template strand ±CF, the block exchange 2 (BE2) labelled on the non-template strand ±CF, the −27 to −23 mutant labelled on the non-template strand ±CF, the −22 to −19 mutant labelled on the non-template strand ±CF, M2, marker for the mutant scaffolds. Arrows point to the band resulting from CF binding on the wt template strand (dark blue), wt non-template strand (cyan), BE1 non-template strand (green) and BE2 non-template strand (purple). The cartoon of transcription scaffolds used for footprinting is shown as lines and the specificity box as circles; BE mutants are shown by green and purple lines. **b** Promoter-dependent in vitro transcription assay was performed using the wt transcription scaffold as in the cryo-EM sample. Top panel: The lanes from left to right show: sample without Rrn3, sample without CF, WT transcription scaffold, BE1, BE2, −29 to −27 mutant, −27 to −23 mutant, −27 mutant, −27 to −26 mutant, −25 mutant, −24 to −23 mutant, −22 to −19 mutant, −14 to −12 mutant, −11 to −7 mutant. Middle panel: The lanes from left to right show: sample without CF, WT transcription scaffold, CF with Rrn7 residues 291–294 mutated to alanine, CF with Rrn7 residues 291–294 DERH mutated to RHDE, CF with Rrn11 with residues R11, K12, K16 and K18 mutated to alanine, CF with Rrn11 with residues R11, K12, K16 and K18 mutated to glutamate, CF with Rrn6 ΔCterm (Δ776–894), sample without Rrn3, Rrn3 Δacidic loop (Δ253–319). Bottom panel: Schematic representation of the template strand of the promoter sequence from −29 to −7 is shown as blue circles. The mutated nucleotides are indicated with the same colour as in the transcription assay. Source data are provided as a Source Data file.

Subsets of particles can be classified into Pol I CC1 and CC2, which accordingly have been depicted as two distinct states in Fig. 6. However, the number of particles is small and the two states result from a large number of classification steps (Supplementary Figs. 1 and 2). We therefore cannot exclude that CC1 and CC2 are rather two snapshots of a continuous movement of the CF-upstream DNA module with respect to Pol I than two distinct CC1 and CC2 states. Time-resolved cryo-EM is just emerging as a new technique to better resolve time-ordered series of conformational changes[37] and could be used to further elucidate this question.

As a next step, small conformational rearrangements of the CF-upstream DNA and Pol I-Rrn3 result in spontaneous DNA opening (Fig. 6c). The transcription bubble is maintained in the cleft by three elements: (1) base flipping, which defines the edge of the bubble, (2) the rudder, which prevents re-annealing of DNA strands and (3) the A49 linker helix, which seals the cleft following clamp closure, maintains the clamp/rudder in a closed state and positions the A49 tWH. Human Pol II also shows similar clamp movements during the transition from CC to OC[2], while yeast Pol II also opens the downstream DNA via distortion of upstream DNA and cleft closure[4].

Once the synthesized RNA reaches a certain length, it clashes with the B-finger of Rrn7 that subsequently clashes with the Rrn3 acidic loop. This clash likely destabilizes the interactions of CF and Rrn3 with Pol I as it moves forward (Fig. 6d) and indeed, deletion of the Rrn3 acidic loop leads to a severe reduction in transcription activity (Fig. 5b). Further transition to the EC requires a slight movement of the upstream DNA that would collide with A49 tWH in the initiation position. As a result, the A49 tWH moves to its elongation position, where it displaces the N-terminal region of Rrn3 to further destabilize Rrn3 binding to Pol I (Fig. 6e). Therefore, transition to the EC and disassembly of the IC after initial RNA synthesis involves both displacement of the N-terminal region of Rrn7 by RNA and destabilization of Rrn3 binding by both movement of the Rrn7 Zn-ribbon and the A49 tWH (Fig. 6f). Our data rationalize the dual role of A49 tWH in IC formation and early elongation[23,25]. First, binding of A49 tWH to the specificity box contributes possibly to DNA opening and maintaining the IC. Second, movement of A49 tWH towards its elongation conformation destabilizes the IC.

The cryo-EM structures of Pol I CCs, intermediates OCs and complete OCs reported here together with previous Pol I IC structures[9–11] provide a working model for the Pol I transcription initiation mechanism. In Pol I, the mechanism of promoter opening appears different compared to Pol III[12,13] although some features are conserved. In the Pol III system, the DNA in the CC is retained out of the clamp by a kink in the downstream DNA[13] similar as in the Pol I CC. The available structures of the Pol III

CCs were assumed to be early engagement intermediates, and it was speculated that in Pol III the transition from the CC to the OC involves transient clamp opening and DNA loading[13]. TFIIIB movement in the two available Pol III CCs is less pronounced compared to the ratcheting movement of CF-upstream DNA with respect to Pol I. The ordering of the TFIIIB 'arms' and the winged-helix domains of the Pol III-specific heterotrimer contribute to Pol III promoter opening and stabilization of the transcription bubble, while in Pol I the insertion of the rudder between template and non-template DNA strands, the gradual closing of the DNA clamp and the ordering of the A49 linker and tWH support these tasks.

While the precise mechanisms are different, there are conceptual parallels between the Pol III and the Pol I systems concerning the extent of contribution of the RNA polymerase to promoter opening. In the carefully regulated Pol II system, dissociable transcription factors provide the DNA-binding domains around the upstream promoter element (i.e. the TATA box) as well as the DNA contacts around the upstream edge of the transcription bubble. In contrast, Pol I and Pol III use dissociable transcription factors for recognition of the upstream promoter element (specificity box/TATA box), but it is the RNA polymerase itself that establishes DNA contacts close to the upstream edge of the transcription bubble (A49 and rudder in Pol I and C34/C82 in Pol III). It was proposed that this presents an adaptation to high initiation frequencies in the Pol III system[13], and here we show that the highly active Pol I system shares this feature. Although our current model comprises all factors required for Pol I basal transcription, future studies will have to include also UAF and TBP to better resemble the cellular context.

## Methods

**Protein purification.** Pol I was purified using a C-terminal TAP tag on the AC40 subunit from *S. cerevisae* strain SC1613. Yeast cells were grown in a 100 L fermenter (Biostat D-DCU; Sartorius) at 30 °C to OD$_{600nm}$ of 6–8. The cell paste was re-suspended in a buffer containing 250 mM Tris-HCl, pH 8, 40% glycerol, 250 mM (NH$_4$)$_2$SO$_4$, 1 mM EDTA, 10 mM MgCl$_2$, 10 µM ZnCl$_2$, 12 mM β-mercaptoethanol in the presence of a protease inhibitor cocktail (complete EDTA-free; Roche) and lysed with glass beads in a BeadBeater (BioSpec). After centrifugation at 14,000 r.p.m. for 1 h at 4 °C, the protein lysate was loaded on heparin-sepharose resin (GE Healthcare). The complex was eluted from the resin using high-salt buffer with 1 M (NH$_4$)$_2$SO$_4$ and further incubated with IgG Sepharose (GE Healthcare) for 6 h. After washing, IgG beads were mixed with tobacco etch virus (TEV) protease and incubated overnight at 4 °C. TAP-tag cleaved Pol I was recovered and subsequently purified by ionic exchange on a Mono-Q column (GE Healthcare). The pure Pol I enzyme was concentrated to 7 mg ml⁻¹ in 15 mM Tris-HCl, pH 8.0, 150 mM NaCl and 10 mM dithiothreitol (DTT). Rrn3 was expressed recombinantly in BL21 (DE3)Star pRARE cells using the pRSF vector (Novagen) in autoinduction (ZY) medium. The temperature was shifted to 20 °C when the cultures reached OD$_{600nm}$ 1.2 and let grow overnight. After cell lysis in 1 M NaCl, 50 mM Tris-HCl pH 8, 20% glycerol, 10 mM MgCl$_2$, DNase, and protease inhibitor

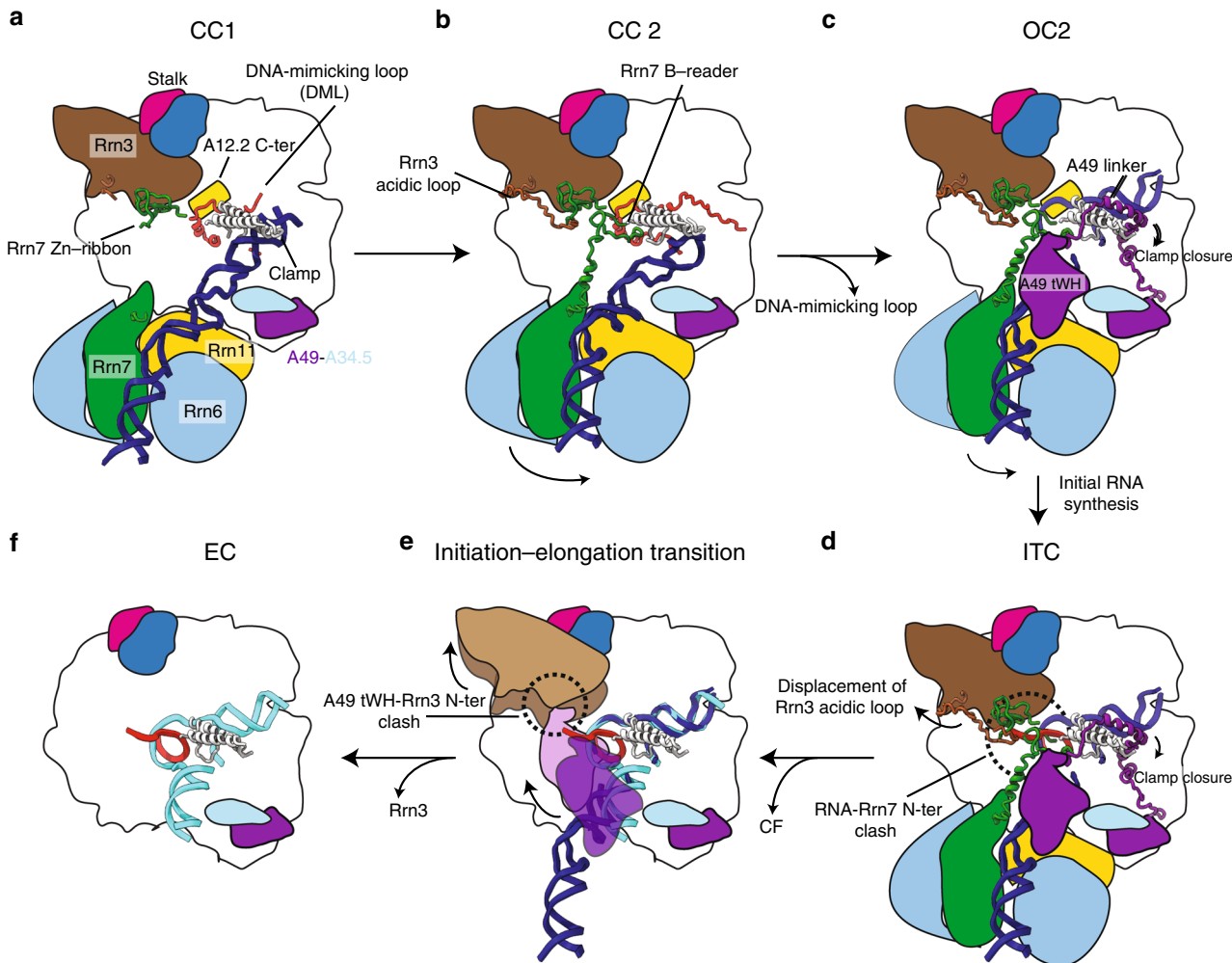

**Fig. 6 Mechanism of Pol I transcription initiation and promoter escape.** Different Pol I transcription states are depicted as cartoon representation of the corresponding structures. Pol I is depicted as a black contour, A12.2 C-terminal domain as a yellow rectangle and the elements involved in the regulation of initiation and transition to elongation are shown in ribbon. In Pol I CC1 (**a**), the CF is bound to upstream DNA in a distal position from Pol I-Rrn3. At this stage, the B-finger of Rrn7 and the acidic loop of Rrn3 are not yet resolved, the cleft is open and the DNA-mimicking loop (red) is present in the active site. Pol I CC1 then transitions to Pol I CC2 (**b**) where CF-upstream DNA rotates with respect to Pol I-Rrn3 in a position closer to its corresponding position in the OC. The acidic loop of Rrn3 and the Rrn7 B-finger become better ordered and presumably stabilize the complex. During DNA opening, the template strand is loaded into the cleft and replaces the DNA-mimicking loop while CF-upstream DNA still rotate slightly to form the OC. Once the clamp closes, the A49 linker wraps around the clamp coiled-coil helices and positions the A49 tWH onto the upstream promoter DNA to form the Pol I OC2 (**c**). Together with CF, the A49 tWH recognizes the specificity box of the DNA promoter. After initial RNA synthesis, the clamp further closes and the ITC forms (**d**). When the RNA reaches a critical length, it clashes with the Rrn7 B-finger, which then displaces the Rrn3 acidic loop. As a result, CF dissociates from the complex. Next, the A49 tWH moves to the EC position where it clashes with Rrn3 and displaces it from the complex (**e**). Finally, RNA synthesis continues and Pol I leaves the promoter and enters the elongation state (**f**).

cocktail, the lysate was incubated for 1 h at 4 °C with Ni-NTA beads (Qiagen). After elution and overnight incubation at 4 °C with TEV protease, the cleaved protein was further purified using a Mono-Q column and Superdex 200 (GE Healthcare) in 150 mM NaCl, 50 mM Tris-HCl pH 8, and 10 mM DTT. Codon optimized genes of CF were expressed in BL21 (DE3) cells in TB medium (Supplementary Fig. 11). Expression was induced using 0.05 mM IPTG at OD$_{600nm}$ 0.8. Cells were harvested after an overnight incubation at 18 °C. Cells were lysed in a buffer containing 500 mM KCl, 50 mM HEPES-NaOH pH 8, 5 mM MgCl$_2$, 0.1 mM EDTA, 10 mM imidazole, 1 mM TCEP, 20% glycerol, DNase I, lysozyme and protease inhibitor cocktail. The lysate was cleared by centrifugation and the supernatant was incubated with Ni-NTA beads for 1 h at 4 °C. The beads were washed twice with a buffer containing 1 M KCl, 50 mM HEPES-NaOH, pH 8, 5 mM MgCl$_2$, 0.1 mM EDTA, 10 mM imidazole, 1 mM TCEP, 20% glycerol supplemented with ATP and further washed twice with a buffer containing 200 mM KCl with ATP and twice with the same buffer without ATP. The protein was eluted in the same buffer supplemented with 300 mM imidazole. The eluent was mixed with TEV protease and incubated at 4 °C overnight. The TEV-cleaved protein was loaded on a Heparin column and a gradient from 200 mM to 1 M KCl was used to elute CF. The protein was further purified using a Superdex 200 column (GE Healthcare) in 150 mM KCl, 20 mM HEPES-NaOH pH 8 and 1 mM TCEP.

**Assembly of ICs.** The OC was prepared by incubation of Pol I at a concentration of 1.5 μM with Rrn3 at 3 μM for 5 h at 4 °C. A transcription scaffold containing the core promoter (−50 to +20) with a 15-bp mismatch (−10 to +5) (Template, 5′-GTCTTCAACTGCTTTCGCATG-AAGTACCTCCCAACTACTTTTCCTCACAC TTGTACTCCATGACTAAACC-3′; non-template, 5′-GGTTTAGTCATGGAGT ACAAGTGTGAGGAAAAGTAGTTGGCGTAGCAGGAGAAGTAAAGCAGT TGAAGAC-3′) was prepared by heating the oligonucleotides at 95 °C for 3 min and cooling down to 25 °C. The transcription scaffold was mixed with Pol I-Rrn3 complex and further incubated for 1 h at 4 °C. Next, CF at a concentration of 1.5 μM was added to the Pol I-Rrn3-DNA complex in buffer A (300 mM potassium acetate, 50 mM HEPES-NaOH pH 8, 10 mM DTT) and dialysed overnight against buffer B (100 mM potassium acetate, 50 mM HEPES-NaOH pH 8, 10 mM DTT). The CC was prepared in two different ways. In the first approach (CC1), 1.5 μM of Pol I was mixed with 3 μM of Rrn3 and incubated overnight at 4 °C. Then, CF at a concentration of 1.5 μM was added to the core promoter (−50 to +20) (Template, the same sequence as used for the OC sample; non-template, 5′- GGGTTT AGTCATGGAGTACAAGTGTGAGGAAAAGTAGTTGGGAGGTACTTCATG CGAAAGCAGTTGAAGAC-3′) and incubated for 1 h at 4 °C. Subsequently, Pol I-Rrn3 complex was added to the CF-DNA in buffer B and used directly for the EM analysis. In the second approach (CC2), Pol I at a concentration of 1.5 μM was

mixed with Rrn3 at 3 μM, CF at 1.5 μM and transcription scaffold at 1.5 μM in buffer B without any incubation time and directly used for plunge freezing.

**Electron microscopy**. Holey copper grids (R 2/1 + 2 nm carbon) (Quantifoil) were glow discharged for 45 s using PELCO easyGlow. The sample (2.5 μl, 0.2–0.3 mg/ml final concentration) was applied on the grids, incubated for 30 s at 4 °C and 100% humidity and blotted for 3 s with blot force 2 using Vitrobot Mark IV (FEI). Data were acquired on a Titan Krios (FEI) at 300 keV through a Gatan Quantum 967 LS energy filter using a 20 eV slit width in zero-loss mode. In all, 8074 movies were recorded for the OC complex on Gatan K2-Summit direct electron detector at a nominal EFTEM (energy-filtered transmission electron microscope) magnification of ×105,000 corresponding to 1.32 Å calibrated pixel size (in 4K mode) (Table 1). The movies were collected in 40 frames in defocus range between −0.75 and −3 μm with a dose of $1.57 e^- Å^{-2}$ per frame. In total, 7338 movies of the CC1 sample were collected at a nominal magnification of ×135,000 corresponding to a pixel size of 1.04 Å. The movies were collected in 40 frames with a dose of $1.05 e^- Å^{-2}$ per frame in defocus range between −0.75 and −3 μm. In all, 21,552 movies were collected for the CC2 sample at a nominal magnification of ×135,000. The movies were collected in 40 frames with a dose of $1.11 e^- Å^{-2}$ per frame in defocus range between −0.5 and −2.5 μm. Data collection was fully automated using SerialEM[38].

**Data processing**. Dose-fractionated movie frames were aligned and dose-weighted using MotionCor2 (ref. [39]). The dose-weighted micrographs were corrected for the contrast transfer function (CTF) using Gctf[40]. The class averages corresponding to Pol I and Pol I PIC[11] were used as a template for autopicking in Relion 2 (ref. [41]). In toal, 1,988,127 particles from the OC dataset, 1,347,851 particles from the CC1 dataset and 5,148,501 particles from the CC2 were picked and sorted by 2D classification to separate ice and low-resolution particles from particles with high-resolution features. Particles showing high-resolution features were selected and refined using the 40 Å low-pass filtered previous Pol I PIC map[11] as reference.

In the OC dataset, 709,034 particles contained a density for CF and were further locally classified on the CF density. The best class (147,227 particles) was further globally classified. The resulting classes of OC1 and OC2 containing 75,851 and 59,963 particles were refined to 4.2 and 3.8 Å resolution, respectively. Focused refinement on the CF-upstream DNA or Pol I-Rrn3-downstream DNA improved the resolution CF-upstream DNA or Pol I-Rrn3-downstream DNA of OC2 to 3.8 and 3.5 Å and of OC1 to 3.9 Å resolution. The remaining particles lacking CF density (214,962 particles) were pooled, refined and 3D classified. Classes that contained a density for A49 tWH (11,631 particles) were pooled and classified. The best class containing 9789 particles was refined to 4.7 Å resolution. The final classes OC1, OC2 and Pol I-Rrn3-tWH were further processed with CtfRefine and Bayesian Polishing pipelines implemented in Relion 3 to obtain improved resolution to 3.5, 3.0 and 4.0 Å, respectively. The OC1 and OC2 were multibody-refined on the CF-upstream DNA and Pol I-Rrn3-downstream DNA densities[42]. The resolution of the CF-upstream DNA and Pol I-Rrn3-downstream DNA densities of the OC1 improved to 3.6 and 3.4 Å and of the OC2 improved to 3.1 and 3.0 Å resolution, respectively.

The high-resolution particles from the CC1 dataset were divided into three groups and classified separately. The majority of classes showed a clear density for CF and classes with the highest number of particles and high-resolution features (501,297 particles) were pooled, refined and classified. Classes with a good density for CF (384,630 particles) were pooled and classified using a mask on Pol I-Rrn3 and the heterodimer (A49-A34.5) densities. The best class with 24,848 particles resulted in the OC intermediate 1 reconstruction at 3.7 Å resolution. One class with a fragmented density for CF (116,667 particles) was separately refined and classified, which resulted in the CC1 reconstruction at 5.4 Å resolution. The CC1 and OC intermediate 1 classes were processed with Relion 3 to 3.8 and 3.1 Å resolution, respectively. Multibody refinement of the CF-DNA and Pol I-Rrn3 of the CC1 resulted in densities at 9.7 and 3.7 Å resolution, respectively. Multibody refinement of the CF-upstream DNA and Pol I-Rrn3-downstream DNA of the OC intermediate 1 improved the resolution to 3.3 and 3.1 Å, respectively.

The CC2 dataset was divided into eight parts and each was separately classified. The resulting classes (139,819 particles) from two parts showed already high-resolution features in the map that were pooled and classified. After a global classification, the best class was further classified using a mask on the heterodimer and further locally classified on either the CF, Pol I-Rrn3 or the heterodimer densities. The particles obtained in the first round of classification which resulted in low-resolution maps were pooled and divided into two independent groups (508,985 and 432,520 particles) that were separately classified. Initial classification allowed sorting particles with a dsDNA density (321,473 particles) from the particles with an open DNA. The OC particles were pooled and further classified based on the Pol I-Rrn3 and the heterodimer densities. The OC particles containing the heterodimer were pooled together (358,983 particles) and classified based on the CF density to obtain a reconstruction of the OC intermediate 2 at 3.5 Å resolution (42,727 particles). The CC particles were classified based on the Pol I-Rrn3, CF, DNA and the heterodimer to obtain the final CC2 reconstruction at 3.8 Å resolution (24,482 particles). The CC2 and OC intermediate 2 classes were processed with Relion 3 to 2.9 and 2.7 Å resolution, respectively. Multibody refinement of the CF-DNA and Pol I-Rrn3 of the CC2 resulted in densities at 3.0 and 2.9 Å resolution, respectively. Multibody refinement of the CF-upstream DNA and Pol I-Rrn3-downstream DNA of the OC intermediate 2 improved the

resolution to 2.8 and 2.7 Å, respectively. All maps were sharpened and amplitude corrected and the local resolution was estimated using Blocres[43].

**Model building**. The models were built by rigid body fitting the Pol I from the Pol I PIC (PDB: 5OA1 [http://dx.doi.org/10.2210/pdb5oa1/pdb]), Rrn3 (PDB: 3TJ1) [http://dx.doi.org/10.2210/pdb3tj1/pdb] and CF (PDB: 5O7X) [http://dx.doi.org/10.2210/pdb5o7x/pdb] into the OC maps and the A49 tWH crystal structure (PDB: 3NFI) [http://dx.doi.org/10.2210/pdb3nfi/pdb] into the Pol I OC2 map. The Pol I crystal structure (PDB: 4C3I) [http://dx.doi.org/10.2210/pdb4c3i/pdb], Rrn3 (PDB: 3TJ1) and CF (PDB: 5O7X) were fitted into the electron densities of the Pol I CC1 and CC2 using Chimera followed by manual building of previously unresolved residues and refinement of the models in COOT[44]. The model of the DNA was used from Pol I ITC (PDB: 5W65) [http://dx.doi.org/10.2210/pdb5w65/pdb] for Pol I OC maps followed by manual building and adjustment in COOT. The downstream DNA in the Pol I CC maps was manually built as an extension to the upstream DNA from the OC models. The final models were real-spaced refined separately for CF, Pol I-Rrn3 and DNA against focused-refined maps and the final models were refined against the full map using Phenix pipeline for cryo-EM maps[45]. The figures were prepared with ChimeraX and Chimera[46,47]. Minor and major grooves widths were calculated with 3DNA web server[48].

**Footprinting and transcription assays**. For exonuclease III footprinting experiments, the DNA template was first labelled at its 5′ end using [γ-$^{32}$P]ATP and T4 PNK. After PAGE purification, the template and non-template strands were paired (one strand is labelled) in 50 mM Tris-HCl pH 7.5, 100 mM potassium acetate and 5 mM MgCl$_2$. In all, 2 pmol of DNA were incubated with 4 pmol of CF for 20 min at room temperature, before addition of 80 units of Exonuclease III (New England Bio-labs) and further incubation for 4 min at 28 °C. The reaction was stopped by addition of 0.1% SDS, phenol extraction and ethanol precipitation. The DNA pellet was then analysed by denaturing 6% PAGE and the radioactivity recorded with a PhosphorImager.

Run-off transcription was performed by mixing 2 pmol of DNA duplex with 4 pmol of CF for 20 min at RT. Then a pre-incubated complex (4 pmol Pol I and 6 pmol Rrn3) was added and further incubated for 10 min. The transcription reaction was initiated by addition of 0.2 mM ATP, 0.2 mM CTP, 0.2 mM GTP and 2.5 μCi α-P$^{32}$ UTP in 40 mM Tris-HCl pH 7.5, 10 mM NaCl, 12 mM MgSO$_4$ and 10 mM DTT. After 15 min at 28 °C, the products were analysed on 17% denaturing PAGE and detected with a PhosphorImager.

**Reporting summary**. Further information on research design is available in the Nature Research Reporting Summary linked to this article.

## Data availability

The coordinates of the atomic models of CC1, CC2, OC1, OC2, Intermediate 1, Intermediate 2 and Pol-Rrn3-tWH have been deposited with the Protein Data Bank under accession code 6RQH, 6RQL, 6RUO, 6RWE, 6RRD, 6RUI, 6RQT. The corresponding cryo-EM maps have been deposited with the Electron Microscopy Data Base under accession code EMD-4982, EMD-4984, EMD-10007, EMD-10038, EMD-4987, EMD-10006 and EMD-4985, respectively. The source data underlying Figs. 5a, b are provided as a Source Data file. Other relevant data are available from the corresponding author upon reasonable request.

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

## Acknowledgements

Y.S., F.B., L.T., R.W. and C.W.M. acknowledge support by the ERC Advanced Grant (ERC-2013-AdG340964-POL1PIC). L.T. acknowledges support by the EMBL International PhD program. The authors thank J. Hanske, M. Vorländer and H. Fung for critically reading the manuscript and fruitful discussions and T. Hoffmann for computational assistance.

## Author contributions

C.W.M. initiated and supervised the project. Y.S. and F.W. collected the cryo-EM data. Y.S. performed the image processing, data analysis and model building and L.T assisted in data analysis and model building. F.B. performed the biochemical experiments. R.W. carried out yeast fermentation and together with B.M. assisted in sample preparation. Y.S, F.B., L.T. and C.W.M. prepared the manuscript with input from all other authors.

## Competing interests

The authors declare no competing interests.
