## [Peer Review File · Nature Communications]

Reviewers' Comments:

Reviewer #1:

Remarks to the Author:

Sadian et al. solved the latest cryo-EM structure of the Pol I PIC. Their new structures detail the closed PIC which has not been observed before as well as higher resolution structures of the open complex. They also reveal some new functions for Rrn3 and A49 during the PIC formation process, although they do not formally biochemically test these observations which would be an excellent addition to this work. They also present a detailed model for Pol I promoter opening, which could be elevated by more discussion of previous findings. Overall, this work will be of general interest to the Pol I field and greater transcription field. This manuscript is suitable for publication in Nature Communications with after addressing the comments detailed below.

Major comments

1. Terse discussion of previous structures. It would be better if the authors more clearly described the difference in the structures as they all present different protein-DNA contacts for CF some larger than others. One could argue that the Han et al. Elife paper showed the CF contacts the most clearly and their structure most closely resembles the ones presented in these higher resolution structures.
2. Promoter opening model – it seems like the same model is being proposed as shown for the Pol III mechanism. Could the authors more clearly discuss this or point to where they are different from Pol III?. Also, could the authors comment on the previous CF cryo-EM papers and their models for promoter opening (Han et al., Elife; Engel et al., Cell). It is unclear if a new model is being presented or it is consistent with these previous models. A discussion of this would be beneficial for the reader.
3. Rrn7 mutations. Could the authors explain more why mutations to Rrn7 have little impact on DNA binding. Is this because it is a structure-based binding versus a sequence based? Is this common for DNA binding proteins? It seems very unsatisfying that a clear contact is presented yet it has no impact on its biological function. Could a similar set of mutations be made in Rrn11 to see if this is a general feature of CF or specific to Rrn7? At the very least either more mutations should be tried and/or richer speculation as to why this phenomenon occurs with Rrn7.
4. Biochemical validation of open complex formation. The paper is lacking some tests of their models. For instance, is A49tWH necessary for open complex formation? For instance, are the residues and contact points important for A49 function? Is the acidic loop of Rrn3 necessary Pol I initiation/early elongation? It would be nice to test their intriguing idea mores to elevate the structural findings. This is especially the case for disordered and mobile domains.

Minor comments

1. Exonuclease mapping. There is room to spell out the words of the DNA strand and proteins. This would be helpful for the reader. Also, why is the exposure so white? It would be nice to see the background on this data as it could show weaker footprints that may correlate better with activity.
2. Movie. The authors should include a movie showing the PIC and its transitions. This may have been missed during the review. Bottom line – this would be a great benefit to the reader and useful for educational purposes.
3. Biochemical validation of CF binding. How do the results present in this paper correlate with the biochemical experiments detailed the Engel et al., Cell paper. Their data may help support the authors' conclusions.

4. Bend and Kinks. Are the bends and kinks of the promoter consistent with the previous Cryo-EM studies?

Reviewer #2:

Remarks to the Author:

The authors of this work describe a very comprehensive cryo electron microscopy study to reveal several stages of the RNA polymerase I pre-initiation complex formation. They analyze [RNA polymerase I - Rrn3- Core Factor] complexes in the presence of rDNA promoter DNA to track a Closed complex and in the presence of a mismatched DNA to induce Open Complex formation. They identify several rare intermediate states, bring important new insights into the role of A49-tWH and of the Core Factor. They discovered a new role for Rrn3 in contacting Core Factor. Altogether this report is of excellent quality and brings new structural insights into the assembly of a RNA Polymerase I PIC.

1- I have one major criticism concerning the Closed Complexes.

The proportion particles included in the reconstruction of some complexes relative to the number of picked particles is dramatically small for the Closed Complexes (less than 1%). On one hand this percentage is misleading since it does not reflect the proportion of OC in solution since many of them were discarded in the first 2D cleaning steps that select for high resolution. On the other hand, with such a small fraction of the particles being in that conformation, can we really be sure that this represents the actual first steps of the reaction or an unrelated possibly nonproductive form of the complex.

More importantly, this proportion did not increase when lowering the incubation time. The CC1 dataset is a long incubation time with more opportunity to evolve towards OC, while the CC2 dataset is identical but with a short incubation time. Although the proportion of closed complexes is the same, the two datasets lead to slightly different 3-D models. This is already surprising since one would expect to reconstruct exactly the same conformation. More surprising is that in the final model the two CC reconstructions are believed to represent two kinetically different stages in RNA Polymerase I PIC formation and in addition CC1 is placed before CC2. Considering the small proportion of particles and the large number of classification steps I would recommend caution in the interpretation of this maps. The authors should comment and draw the attention of the reader to the limits of this interpretation.

2- When partitioning the OC dataset, the authors identified a new position of the A49-tWH domain (OC2). The authors should comment on the difference between the OC1 and OC2 dataset which makes the A49-tWH domain adopt the TFIIE β position only in OC2. It was not fully clear for me, whether this position also found when analyzing the spontaneously formed OCs. This could be recalled in the text if true, if not the authors need to comment.

3- A number of small details need to be corrected:

- Line 80: Check the first sentence of the results section
- The contraction "IC" not defined
- Line 104 The authors write "the downstream DNA adopts an unexpected position by being strongly bent away from the DNA-binding cleft at bp -11 to -7". If upstream and downstream are defined according to the TSS, then the downstream DNA (+1 to +20) is not seen in the map, and the bending occurs in the upstream part. In this respect indicating the position of the TSS would be useful in fig.1b.
- Line 136 I am not convinced that reference 20 shows that Rrn3 is required in mammalian system
- Line 138 The authors should also mention ref 6
- For sake of clarity, the author should recall which subunit forms the "clamp coiled-coil" domain and that it is located at the basis of the clamp.
- Line 232 check whether IC2 is correctly used
- Line 262 why radiolabelled closed core promoter DNA

- Supplemental material: Check consistency of particle numbers in the supplemental figures and in the material and methods section (for example OC in text 1.639 k particles while 1.988 k in the supplemental figure 3)

Point-by-Point Response

Reviewer #1

Major comments:

1) Terse discussion of previous structures. It would be better if the authors more clearly described the difference in the structures as they all present different protein-DNA contacts for CF some larger than others. One could argue that the Han et al. Elife paper showed the CF contacts the most clearly and their structure most closely resembles the ones presented in these higher resolution structures.

Response: We now describe more clearly the differences to the previously reported Pol I initiation complex (IC) structures. The three recent Pol I IC structures show similar overall architectures, similar DNA conformations and CF-DNA interactions. Differences comprise the presence of RNA in one IC reconstruction (Engel et al, 2017), the presence of the A12.2 C-terminal domain in the active site concomitant with a slightly more open DNA binding cleft in another IC reconstruction (Sadian et al, 2017) and finally the absence of Rrn3 in the third reconstruction (Han et al., 2017). A recent review (Engel et al., 2018) provides a detailed comparison and is now cited in the manuscript.

The effective resolution of the different IC reconstructions considerably varies especially in the CF-DNA interface, but in none of them the resolution allowed assigning the register of DNA-contacting amino acid residues. Nevertheless, we agree with the reviewer that the reconstruction of Han et al., 2017 resembles most our current higher resolution reconstructions except of the presence of the A49 tWH in our structural model. This group also solved an IC structure using a truncated nucleic acid scaffold that allowed them to define the register of the upstream promoter DNA sequence, while they presented only helices and loops that interact with the upstream promoter without being able to pinpoint to specific residues. In contrast, the higher resolution of our reconstruction allowed us to resolve the register of the residues that interact with specific nucleotides as a basis for further mutational studies. To address this comment of the reviewer, we have added the following sections in the introduction, page 3 and at page 11 of the revised manuscript:

Page 3, lines 58-64: *“In comparison, the three Pol I IC structures show similar overall architectures, similar DNA conformations and CF-DNA interactions⁹⁻¹¹. Differences include the presence of RNA in one IC reconstruction⁹, the presence of the A12.2 C-terminal domain in the active site concomitant with a slightly more open DNA binding cleft in another IC reconstruction¹¹ and finally the absence of Rrn3 in the third reconstruction¹⁰. A more detailed comparison is provided in a recent review¹⁴.”*

Page 11, lines 245-249: *“Interaction of CF with DNA is mediated via several elements of the N- and C-terminal cyclin folds of Rrn7, and the TPR protein Rrn11, most of which were previously shown to be involved in DNA interaction and complex formation¹⁰. However, the higher resolution of the OC2 reconstruction allowed us identifying the register of the amino acid residues that specifically interact with promoter DNA as a basis for subsequent mutational studies.”*

2) Promoter opening model – it seems like the same model is being proposed as shown for the Pol III mechanism. Could the authors more clearly discuss this or point to where they are different from Pol III?. Also, could the authors comment on the previous CF cryo-EM papers and their models for promoter opening (Han et al., Elife; Engel et al., Cell). It is unclear if a new model is being presented or it is consistent with these previous models. A discussion of this would be beneficial for the reader.

Response: The mechanism of promoter opening appears different in Pol I compared to Pol III as suggested by Vorländer et al., although some features are conserved. In the Pol III system, the DNA in the closed complex (CC) is retained out of the clamp by a kink in the downstream DNA similar as in the Pol I CC. Transition from the CC to the OC involves transient clamp opening and DNA loading, while transcription factor TFIIB only gradually changes its position with respect to Pol III compared to the more pronounced ratcheting movement of CF/upstream DNA with respect to Pol I. In Pol III, ordering of TFIIB and the winged-helix domains of the Pol III-specific heterotrimer contribute to promoter opening and stabilization of the transcription bubble, whereas in Pol I gradual closing of the DNA clamp and ordering of the A49 linker and the tandem winged-helix domain support these tasks. The ‘ratcheting’ mechanism of CF/upstream DNA was initially suggested by Han et al., 2017 based on different OC conformations, which we now complemented by CCs further corroborating this mechanism.

We now expand on the ratcheting mechanism on page 14, lines 323-331:

“Our structures illustrate the major structural rearrangements that result in spontaneous DNA opening and OC formation during Pol I transcription initiation thereby further completing the ‘ratcheting’ model proposed for Pol I transcription initiation¹⁰. In this model, movement of CF and CF-bound upstream DNA relative to Pol I generates strain on the DNA, which contributes to DNA melting. The ratcheting mechanism of CF-upstream DNA was initially suggested based on different OC conformations¹⁰ and these were observed only in absence of the essential factor Rrn3. We now demonstrate that the ratcheting movement also occurs in the presence of all essential transcription factors and during the transition from the CCs to OCs, thereby further clarifying and corroborating this mechanism.”

We also compare the mechanisms of promoter opening by Pol I and Pol III in the last paragraph of the discussion at page 16, lines 379-402:

“The cryo-EM structures of Pol I CCs, intermediates OCs and complete OCs reported here together with previous Pol I IC structures⁹⁻¹¹ provide a working model for the Pol I transcription initiation mechanism. In Pol I, the mechanism of promoter opening appears different compared to Pol III^{12,13} although some features are conserved. In the Pol III system, the DNA in the closed complex (CC) is retained out of the clamp by a kink in the downstream DNA¹³ similar as in the Pol I CC. The available structures of the Pol III CCs were assumed to be early engagement intermediates, and it was speculated that in Pol III the transition from the CC to the OC involves transient clamp opening and DNA loading¹³. TFIIB movement in the two available Pol III CCs is less pronounced compared to the ratcheting movement of CF-upstream DNA with respect to Pol I. The ordering of the TFIIB ‘arms’ and the winged-helix domains of the Pol III-specific heterotrimer contribute to Pol III promoter opening and stabilization of the transcription bubble, while in Pol I the insertion of the rudder between template and non-template DNA strands, the gradual closing of the DNA clamp and the ordering of the A49 linker and tWH support these tasks.

While the precise mechanisms are different, there are conceptual parallels between the Pol III and the Pol I systems concerning the extent of contribution of the RNA polymerase to promoter opening. In the carefully regulated Pol II system, dissociable transcription factors provide the DNA-binding domains around the upstream promoter element (i.e. the TATA box) as well as the DNA contacts around the upstream edge of the transcription bubble. In contrast, Pol I and Pol III use dissociable transcription factors for recognition of the upstream promoter element (specificity box/TATA box), but it is the RNA polymerase itself that establishes DNA contacts close to the upstream edge of the transcription bubble (A49 and rudder in Pol I and C34/C82 in Pol III). It was proposed that this presents an adaptation to high initiation frequencies in the Pol III system¹³, and here we show that the highly active Pol I system shares this feature.”

3) Rrn7 mutations. Could the authors explain more why mutations to Rrn7 have little impact on DNA binding. Is this because it is a structure-based binding versus a sequence based? Is this common for DNA binding proteins? It seems very unsatisfying that a clear contact is presented yet it has no impact on its biological function. Could a similar set of mutations be made in Rrn11 to see if this is a general feature of CF or specific to Rrn7? At the very least either more mutations should be tried and/or richer speculation as to why this phenomenon occurs with Rrn7.

Response: Our initial experiments showed indeed that mutating DNA-contacting residues in Rrn7 to alanine did not have a pronounced effect on the Pol I transcriptional activity. Like the reviewer, we have been puzzled by this observation and repeated our experiments, but also tested additional Rrn7 and Rrn11 mutations. Our new experiments show indeed that mutating DNA-contacting loop D291-E292-R293-H294 into alanine results in a moderate, but significant, reduction of Pol I transcriptional activity, while the charge reversal of this loop to R291-H292-D293-E294 leads to further reduction of the transcriptional activity. Yet another CF mutant in which Rrn7 residues 293, 294 and 297 were all changed into alanine or glutamate also showed a severe reduction of Pol I transcriptional activity, but has not been included. The discrepancy with our previous experiments are likely caused by differences in the measured protein concentration. Moreover, we have constructed additional CF mutants, where Rrn11 residues R11, K12, R16 and K18 have been changed into alanine or glutamate which strongly affected the Pol I transcriptional activity. The new results have now been included into Fig. 5b of the revised manuscript and on page 13, lines 301-307 of the revised manuscript:

“When we mutated Rrn7 residues 291-294 that directly contact bp -27/-26 all into alanine, we observed a reduced transcriptional activity that was even more pronounced when the charges were reversed (DERH into RHDE). Similarly in Rrn11, when we changed residues R11, K12, K16 and K18 all into alanine or glutamate, we also observed a severe decrease in Pol I transcriptional activity, showing the importance of these residues in Pol I promoter recognition (Fig. 5b, middle panel).”

4) Biochemical validation of open complex formation. The paper is lacking some tests of their models. For instance, is A49tWH necessary for open complex formation? For instance, are the residues and

contact points important for A49 function?) Is the acidic loop of Rrn3 necessary Pol I initiation/early elongation? It would be nice to test their intriguing ideas more to elevate the structural findings. This is especially the case for disordered and mobile domains.

Response: The role of A49 tWH during both initiation and elongation has been previously investigated. The A49 tWH was shown to be important for promoter-dependent *in vitro* transcription (Pilsl et al., 2016) and *in vivo* it was shown to be important for binding and release of Rrn3 (Beckouet et al., 2008). In addition, Geiger et al., 2010 showed binding of purified A49 tWH to Pol I promoter *in vitro* and identified the residues necessary for binding. In contrast, so far the role of the A49 tWH in open complex formation and the role of their DNA-contacting residues in the context of Pol I and the Pol I PIC has not been explored. As we use endogenous yeast Pol I for our studies such experiments would require purifying mutated yeast Pol I or using the yeast strain Y2670 harboring Pol I Δ rap49 (Pilsl et al., 2016) to reconstitute full-length Pol I with recombinant expressed mutated A49/A34.5 heterodimer (point mutations or truncations). Although such experiments are in principle possible, they are time consuming and the success is difficult to predict (see also Tafur et al., 2019). We have therefore not pursued the mutational analysis of the A49 tWH and its role in complex formation. Instead, we have been focusing on subunit Rrn3, where we now show that deletion of the Rrn3 acidic loop (Δ 253-319) results in a significant reduction of the transcriptional activity compared to wt Rrn3. In addition, we also tested truncated Rrn6 in an *in vitro* transcription assay where the C-terminal moiety of Rrn6 (Δ 776-894), not visible in the cryo-EM structure, is lacking. Compared to wt Rrn6, deleting the C-terminal moiety did not affect the *in vitro* Pol I transcription activity. Our results demonstrate the importance of the Rrn3 acidic loop in transcription initiation, which might result from interaction with the A49 linker and tWH and possibly also with the Rrn7 Zn-ribbon. In contrast, there is no indication that the C-terminal end of Rrn6 is involved in Pol I PIC assembly, as deleting it does not affect Pol I transcriptional activity. The results of these new additional experiments are now described at page 13/14, lines 308-318 of the revised manuscript:

*“The C-terminal moiety of Rrn6 is not visible in the EM structure. To track its putative implication in transcription initiation, a CF mutant where the C-terminal moiety of Rrn6 (Δ 776-894) is lacking was used in an *in vitro* transcription assay. Compared to the wt CF, no change was observed in the *in vitro* Pol I activity (Fig. 5b) suggesting that this domain is not involved in CF/DNA recognition and initiation. Deletion of the Rrn3 acidic loop (Δ 253-319) resulted in a significant reduction of transcription compared to wt Rrn3 (Fig. 5b, middle panel). This highlights its functional role during Pol I initiation/early transcription. When the RNA reaches a critical length, it clashes with the Rrn7 B-finger, which then displaces the Rrn3 acidic loop. The absence of the acidic loop presumably prevents CF to dissociate from the complex and leads to poor transcription efficiency.”*

Minor comments

1) Exonuclease mapping. There is room to spell out the words of the DNA strand and proteins. This would be helpful for the reader. Also, why is the exposure so white? It would be nice to see the background on this data as it could show weaker footprints that may correlate better with activity.

Response: In Fig. 5a, we now replaced the gel picture showing the EcoIII footprinting assay with a gel picture that has been exposed longer. In addition, in Fig. 5b, we now separate panels for DNA and protein mutants and spell out “Template DNA” and “Non-template DNA”.

2) Movie. The authors should include a movie showing the PIC and its transitions. This may have been missed during the review. Bottom line – this would be a great benefit to the reader and useful for educational purposes.

Response: In the revised version of the manuscript, we now included a movie (CC1_OC2_transition that illustrate the transitions from CC1 to OC2.

3) Biochemical validation of CF binding. How do the results present in this paper correlate with the biochemical experiments detailed the Engel et al., Cell paper. Their data may help support the authors' conclusions.

Response: Engel et al. deleted or randomized 10 bp regions in the promoter for *in vitro* transcription assay and they observed that deletion/randomization of the region -30 to +3 abolishes transcription. However, they did not provide a more detailed picture which nucleotides/residues are critical for transcriptional activity and CF binding. Our results are in agreement with their findings and we have added two sentences at page 12, lines 270-272:

“It was previously shown that 10 bp sequence randomization in the -30 to +3 region abolishes promoter-dependent transcription in vitro⁹. However, the sequence specificity of CF binding and the critical bases for transcription were not characterized.”

4) Bend and Kinks. Are the bends and kinks of the promoter consistent with the previous Cryo-EM studies?

Response: The EM density of the promoter DNA in the reconstruction of Engel et al., 2017 had a low local resolution and the model of their promoter DNA is considerably shorter in the upstream direction. Therefore, it does not allow a direct comparison. The reconstruction of the promoter region by Han et al., 2017 shows very similar bends and kinks as our OC reconstructions and overall superimposes very well. Our reconstructions of the previously not reported CCs contain an additional bend that is specific to the CCs.

Reviewer #2

1) I have one major criticism concerning the Closed Complexes. The proportion particles included in the reconstruction of some complexes relative to the number of picked particles is dramatically small for the Closed Complexes (less than 1%). On one hand this percentage is misleading since it does not reflect the proportion of OC in solution since many of them were discarded in the first 2D cleaning steps that select for high resolution. On the other hand, with such a small fraction of the particles being in that conformation, can we really be sure that this represents the actual first steps of the reaction or an unrelated possibly nonproductive form of the complex. More importantly, this proportion did not increase when lowering the incubation time. The CC1 dataset is a long incubation time with more opportunity to evolve towards OC, while the CC2 dataset is identical but with a short incubation time. Although the proportion of closed complexes is the same, the two datasets lead to slightly different 3-D models. This is already surprising since one would expect to reconstruct exactly the same conformation. More surprising is that in the final model the two CC reconstructions are believed to represent two kinetically different stages in RNA polymerase I PIC formation and in addition CC1 is placed before CC2. Considering the small proportion of particles and the large number of classification steps, I would recommend caution in the interpretation of this maps. The authors should comment and draw the attention of the reader to the limits of this interpretation.

Response: We agree with the recommendation of the reviewer to be more careful in interpreting the different stages observed in Pol I PIC formation. When preparing the CCs, we mixed CF, DNA and Pol I-Rrn3 followed by 1 h or without incubation time for the CC1 and CC2 dataset, respectively. However even for CC2, the time between complex formation and plunge freezing is still long compared to the time span required by Pol I for opening the DNA. Accordingly, most of the complexes observed on the EM grid represented OCs. In addition, we also obtained intermediate structures between CC1, CC2 and the OC reconstructions. Nevertheless, we were able to classify (admittedly small) subsets of particles to obtain CC1 and CC2 reconstructions with good map quality and interpretability. We note that it is not uncommon in cryo-EM studies that a large fraction of particles is discarded during processing, even if less extensive classification into different functional states is undertaken. This is a result of the EM sample preparation procedure (see for example D'Imprima E et al, eLife, 2019), and we do not claim that the percentage of retained particles in the final classes represents the situation in solution.

Based on the ratcheting mechanism of CF-upstream DNA with respect to Pol I suggested by Han et al. 2017, we have placed the CC1 reconstruction before CC2 because in CC1 the DNA axis is rotated out by about 15 degree compared to CC2 where the DNA adopts a position closer to the position in the OCs. Our model does not take account of the different incubation times prior to plunge freezing for CC1 and CC2 as we consider our freezing protocol not to be reproducible enough to deduce any time dependence.

However, we cannot rule that CC1 and CC2 correspond to two snapshots of a more continuous movement of the CF-upstream DNA module with respect to Pol I. In contrast, we think it is unlikely that CC1 is an unproductive complex and also we cannot exclude it, we don't see any evidence for this hypothesis.

Accordingly, we now state in the revised manuscript at page 15, 351-357:

“Subsets of particles can be classified into Pol I CC1 and CC2, which accordingly have been depicted as two distinct states in Figure 6. However, the number of particles is small and the two states result from a large number of classification steps (Supplementary Fig. 1 and 2). We therefore cannot exclude that CC1 and CC2 are rather two snapshots of a continuous movement of the CF-upstream DNA module with respect to Pol I than two distinct CC1 and CC2 states. Time-resolved cryo-EM is just emerging as a new technique to better resolve time-ordered series of conformational changes³⁵ and could be used to further elucidate this question”

2) When partitioning the OC dataset, the authors identified a new position of the A49-tWH domain (OC2). The authors should comment on the difference between the OC1 and OC2 dataset which makes the A49-tWH domain adopt the TFIIE β position only in OC2. It was not fully clear for me, whether this position also found when analyzing the spontaneously formed OCs. This could be recalled in the text if true, if not the authors need to comment.

Response: We have collected only one dataset with an open transcription scaffold which resulted in OC1 and OC2 reconstructions that differed in the clamp conformation (more closed in the OC2) and the presence of A49 tWH in the OC2 (described in more detail at page 7/8). We also observed exactly the same conformation of the A49 tWH in the spontaneously opened complex but we did not present it in the paper to avoid repetition. We have now mentioned this point at page 7/8, 166-165:

“Interestingly, this novel, TFIIE-like position of the A49 tWH (also observed in the spontaneously formed OC) differs from its position in the EC structure²⁰ and the EC-like position in the Pol I ITC lacking Rrn3¹⁰.”

3) A number of small details need to be corrected:

• Line 80: Check the first sentence of the results section. The contraction “IC” is not defined.

Response: The abbreviation for initiation complex (IC) was introduced in line 56 of the original manuscript. However, for clarity and better readability, we now spell it out again in the first line of the result section.

• Line 104: The authors write *“the downstream DNA adopts an unexpected position by being strongly bent away from the DNA-binding cleft at bp -11 to -7”*. If upstream and downstream are defined according to the TSS, then the downstream DNA (+1 to +20) is not seen in the map, and the bending occurs in the upstream part. In this respect, indicating the position of the TSS would be useful in Fig. 1b.

Response: We agree with the reviewer that the term “downstream DNA” is here misleading. Accordingly, we have changed this sentence into:

“the upstream DNA preceding the TSS adopts an unexpected position by being strongly bent away from the DNA-binding cleft at bp -11 to -7”

We have also indicated position +1 corresponding to the TSS in Fig. 1b as suggested by the reviewer.

- Line 136: I am not convinced that reference 20 shows that Rrn3 is required in mammalian system

Response: The reviewer is right and we removed the reference Peyroche et al., 2000 from line 151.

- Line 138 The authors should also mention ref 6

Response: We now also cite ref 6 (Pils1 et al., 2016) at this position.

- For sake of clarity, the author should recall which subunit forms the “clamp coiled-coil” domain and that it is located at the basis of the clamp.

Response: We have added this additional information at page 6, lines 127-130:

“The open DNA-binding cleft allows Pol I to accommodate downstream dsDNA between the clamp and the lobe where it is maintained by interactions with the protrusion, rudder and the clamp coiled-coil that is formed by the largest subunit A190 at the basis of the clamp (A190 residues 380-470).”

- Line 232 check whether IC2 is correctly used

Response: We have corrected IC2 to CC2 in the revised manuscript. We thank the referee for pointing us to this mistake.

- Line 262 why radiolabelled closed core promoter DNA

Response: We have changed this sentence into:

“We therefore explored the specificity of CF binding using Exonuclease III (ExoIII) footprinting experiments with radiolabelled double-stranded core promoter DNA (bp -50 to +20).”

- Supplemental material: Check consistency of particle numbers in the supplemental figures and in the material and methods section (for example OC in text 1.639 k particles while 1.988 k in the supplemental figure 3)

Response: We have corrected this mistake and changed 1,639,745 to 1,988,127 particles. In addition, we have checked the consistency of particle numbers throughout the manuscript.

Reviewers' Comments:

Reviewer #1:

Remarks to the Author:

The authors addressed all the comments for both reviewers in great detail which included new data and additional text to help the reader. This is beautiful and important work on the Pol I transcription system. I have two minor comments that could be addressed to elevate the paper a bit more.

Some additional minor revisions could be incorporated in the final version. It could be noted that Rrn6 C-terminal domain is non-essential in yeast growth assays which fits well the new results in the revised manuscript. See reference below.

Knutson BA, Luo J, Ranish J, Hahn S. Architecture of the *Saccharomyces cerevisiae* RNA polymerase I Core Factor complex. *Nat Struct Mol Biol.* 2014 Sep;21(9):810-6.

Also a recent publication characterizing CF's DNA binding properties could be discussed as well and it agrees well and complements the results presented in the revised manuscript. See reference below.

Jackobel AJ, Zeberl BJ, Glover DM, Fakhouri AM, Knutson BA. DNA binding preferences of *S. cerevisiae* RNA polymerase I Core Factor reveal a preference for the GC-minor groove and a conserved binding mechanism. *Biochim Biophys Acta Gene Regul Mech.* 2019 Sep;1862(9):194408.

Point-by-Point Response

Reviewer #1:

The authors addressed all the comments for both reviewers in great detail which included new data and additional text to help the reader. This is beautiful and important work on the Pol I transcription system. I have two minor comments that could be addressed to elevate the paper a bit more.

Some additional minor revisions could be incorporated in the final version. It could be noted that Rrn6 C-terminal domain is non-essential in yeast growth assays which fits well the new results in the revised manuscript. See reference below.

Knutson BA, Luo J, Ranish J, Hahn S. Architecture of the *Saccharomyces cerevisiae* RNA polymerase I Core Factor complex. *Nat Struct Mol Biol.* 2014 Sep;21(9):810-6.

Also a recent publication characterizing CF's DNA binding properties could be discussed as well and it agrees well and complements the results presented in the revised manuscript. See reference below.

Jackobel AJ, Zeberl BJ, Glover DM, Fakhouri AM, Knutson BA. DNA binding preferences of *S. cerevisiae* RNA polymerase I Core Factor reveal a preference for the GC-minor groove and a conserved binding mechanism. *Biochim Biophys Acta Gene Regul Mech.* 2019 Sep;1862(9):194408.

Response: Following the recommendation of this referee we now included these two additional references and discuss them at page 12 and page 13 of the revised manuscript.

Page 12: *“The observed CF-DNA interactions and footprinting experiments (Figs. 4i & 5a) are overall in good agreement with the results of Jackobel et al. who used DNA competitor oligonucleotides to map the minimal, sequence-specific CF binding region from bp -28 to bp -17³⁵.”*

Page 13: *Compared to the wt CF, no change was observed in the in vitro Pol I activity (Fig. 5b) suggesting that this domain is not involved in CF/DNA recognition and initiation, which agrees with the fact that deletion of the C-terminal domain of Rrn6 did not affect yeast growth³⁶.*